



# CH$_4$ and N$_2$O fluctuations during the penultimate deglaciation

Loïc Schmidely[1], Christoph Nehrbass-Ahles[2], Jochen Schmitt[1], Juhyeong Han[1], Lucas Silva[1], Jinwha Shin[3,a], Fortunat Joos[1], Jérôme Chappellaz[3], Hubertus Fischer[1], and Thomas F. Stocker[1]

[1]Climate and Environmental Physics, Physics Institute and Oeschger Centre for Climate Change Research, University of Bern, Bern 3012, Switzerland
[2]Department of Earth Sciences, University of Cambridge, Cambridge, UK
[3]CNRS, Univ. Grenoble-Alpes, Institut des Géosciences de l'Environnement (IGE), Grenoble, France
[a]Present address: Department of Earth and Atmospheric Sciences, University of Alberta, Edmonton, AB, T6G 2E3, Canada

**Correspondence:** Loïc Schmidely (loic.schmidely@climate.unibe.ch)

**Abstract.** Deglaciations are characterized by the largest natural changes in methane (CH$_4$) and nitrous oxide (N$_2$O) concentrations of the past 800 thousand years. Reconstructions of millennial to centennial-scale variability within these periods are mostly restricted to the last deglaciation. In this study, we present composite records of CH$_4$ and N$_2$O concentrations from the EPICA Dome C ice core covering the penultimate deglaciation at temporal resolutions of ∼100 years. Our data permit
the identification of centennial-scale fluctuations standing out of the overall transition to interglacial levels. These features occurred in concert with reinvigorations of the Atlantic Meridional Overturning Circulation (AMOC) and northward shifts of the Intertropical Convergence Zone. The abrupt CH$_4$ and N$_2$O rises at ∼134 and ∼128 thousand of years before present (hereafter ka BP) are assimilated to the fluctuations accompanying the Dansgaard–Oeschger events of the last glacial period, while rising N$_2$O levels at ∼130.5 ka BP are assimilated to a pattern of increasing N$_2$O concentrations that characterized
the end of Heinrich stadials. We suggest the 130.5-ka event to be driven by a partial reinvigoration of the AMOC. Overall, the CH$_4$ and N$_2$O fluctuations during the penultimate deglaciation exhibit modes of variability that are also found during the last deglaciation. However, trace gas responses may differ for similar type of climatic events, as exemplified by the reduced amplitude and duration of the 134-ka event compared to the fluctuations of the Bølling–Allerød during the last deglaciation.

## 1 Introduction

Methane (CH$_4$) and nitrous oxide (N$_2$O) are the second and third most potent well-mixed gases in terms of radiative forcing ($0.48 \pm 0.05$ Wm$^{-2}$ and $0.17 \pm 0.03$ Wm$^{-2}$, respectively) after carbon dioxide (CO$_2$) (Myhre et al., 2013). The impact of these trace gases on the Earth's radiative balance in the future depends also on the sensitivity of natural sources to anthropogenic warming. Time periods of climate change in the past provide natural templates to study this coupling (Fischer et al., 2018). Reconstructions of trace gas concentrations before the instrumental era are only enabled by analyzing the composition of air
trapped in tiny bubbles in polar ice cores, reflecting the atmospheric composition at the time the bubbles were formed. Ice core records of CH$_4$ and N$_2$O concentrations combined with temperature reconstructions documented the natural variability of the trace gases and their coupling to climate change during the glacial cycles of the past 800 thousand years. The overall increase in concentrations accompanying deglaciations represent the largest recurring changes (Spahni et al., 2005; Loulergue et al.,



2008; Schilt et al., 2010a). Records spanning the last deglaciation (Termination I (TI), 18–11 thousand of years before present
(hereafter ka BP), where *present* is defined as 1950 Common Era) showed that this overall increase appears as a sequence
of millennial and centennial fluctuations superposed, for $CH_4$, on longer-term gradually rising concentrations (Marcott et al.,
2014; Rhodes et al., 2015; Fischer et al., 2019). Records resolving short-term fluctuations within deglaciations are limited to
TI, owing to the availability of multiple high-accumulation ice cores.

The aim of this study is to extend the deglacial record to the penultimate deglaciation (Termination II (TII), 140–128 ka BP).
We present high-resolution $CH_4$ and $N_2O$ composite datasets from the EPICA Dome C (EDC) ice core including 150 new
measurements covering the time interval from 145–125 ka BP, combined with the published data of Loulergue et al. (2008)
and Schilt et al. (2010a). We increased the sampling density of the aforementioned records by a factor ∼3.5 and ∼5 to obtain
mean resolutions of 100 and 115 years for $CH_4$ and $N_2O$, respectively, on the order of the mean width of the gas age distribution
(GAD) at EDC (estimated at ∼170 years for TII using the approach of Nehrbass-Ahles et al. (2020)). In addition, we present
a coarse resolution record of $N_2O$ isotopic composition ($\delta^{15}N(N_2O)$ and $\delta^{18}O(N_2O)$), used to assess the integrity of the $N_2O$
measurements. Overall, our data allow us to study the extent to which the evolution of $CH_4$ and $N_2O$ concentrations differ
between the last two deglaciations.

Millennial fluctuations observed within TI belong to a mode of variability, most notably exemplified in the frame of the
Dansgaard–Oeschger (DO) events of the last glacial, where $CH_4$ and $N_2O$ concentrations respond to Greenland temperature
fluctuations. The DO-like mode of variability is characterized by abrupt increases in $CH_4$ concentrations (∼50–260 ppb in a
few centuries), synchronous or slightly lagging the onsets of interstadial Greenland warming by a few decades (Baumgartner
et al., 2014; Huber et al., 2006; Rosen et al., 2014), while $N_2O$ concentrations exhibit concomitant fluctuations reaching up
to ∼60 ppb (Flückiger et al., 2004; Schilt et al., 2010a, 2013). In addition, particularly low $N_2O$ concentrations are observed
during Heinrich stadials (HS), extended stadials defined by the occurrence of massive iceberg discharges through Hudson Strait
into the North-Atlantic (Hemming, 2004). At the end of the HS, $N_2O$ concentrations start increasing centuries to millennia
before the Greenland temperature and $CH_4$ rises (Schilt et al., 2013) (*Late stadial* increase). This mode of variability has been
evidenced for the HS during the last glacial period (Schilt et al., 2013) and the last deglaciation (Fischer et al., 2019; Schilt
et al., 2014).

The centennial $CH_4$ fluctuations within TI belong to a mode of variability found in HS 1,2,4 and 5 (*Intra-stadial* variability).
This pattern consists of short-lived increases reaching amplitudes of ∼30–55 ppb and characteristic timescales of ∼100–300
years, associated with large iceberg discharges in the North-Atlantic (Rhodes et al., 2015).

DO-like and intra-stadial $CH_4$ fluctuations are likely driven by changes in tropical wetland emissions (Rhodes et al., 2015;
Bock et al., 2017), where $CH_4$ is produced by the decomposition of organic matter under anaerobic conditions. Changes in
geologic and pyrogenic emissions as well as changes in the sink strength play only a minor role (Dyonisius et al., 2020; Bock
et al., 2010, 2017; Levine et al., 2012; Hopcroft et al., 2017). Wetland emissions are controlled by climate (precipitation,
temperature, and atmospheric $CO_2$ concentration), modulating wetland extent, emission rates, and ecosystem composition
(Van Groenigen et al., 2011; Melton et al., 2013; Bloom et al., 2010). Changes in tropical wetland emissions during DO-like
events are linked to the strengthening of monsoonal precipitation in the Northern Hemisphere (NH) tropics, enhancing wetland



emission rates (Bock et al., 2017). Increased NH tropical precipitation is associated to northward shifts of the Intertropical

Convergence Zone (ITCZ) in response to changes in heat distribution by the Atlantic Meridional Overturning Circulation (AMOC) (Broccoli et al., 2006; Alley, 2007). On the other hand, intra-stadial fluctuations are believed to result from southward shifts of the ITCZ, strengthening monsoonal precipitation in the Southern Hemisphere (SH) tropics, leading to an increase in wetland emissions there (Rhodes et al., 2015).

DO-like $N_2O$ variability is likely driven by changes in emission from the terrestrial and marine biospheres, where $N_2O$

is emitted as a by-product of nitrification and intermediate product of denitrification (Joos et al., 2019, 2020; Fischer et al., 2019; Schilt et al., 2014). Terrestrial emissions are controlled by climate (precipitation, temperature, and atmospheric $CO_2$ concentration) and available land area (Joos et al., 2020; Van Groenigen et al., 2011). During TI, the response of terrestrial sources to DO-like fluctuations is believed to result from temperature and precipitation changes (Joos et al., 2020), appeared in phase with Greenland warming, and lasted maximum ∼200 years (Fischer et al., 2019). Marine emissions are linked to

the strength of the AMOC, modulating oxygen concentrations in the upper ocean and the amount of organic matter converted into minerals at depth. During DO-like events, marine emissions are believed to be stimulated mainly by deoxygenation in the upper ocean as a consequence of the reinvigoration of the AMOC (Joos et al., 2019). Finally, the late stadial $N_2O$ increases during TI is driven exclusively by marine emissions (Fischer et al., 2019; Schilt et al., 2014), maybe resulting from a long-term reorganization of the nitrate and oxygen concentrations following the preceding AMOC collapse (Schmittner and Galbraith,

75 2008).

## 2 Methods

The results presented in this study are derived from two different instruments. The $\delta^{15}N(N_2O)$ and $\delta^{18}O(N_2O)$ data were measured with the device described in Schmitt et al. (2014), while $CH_4$ and $N_2O$ concentrations were performed with a newly developed analytical system, firstly deployed for this measurement campaign. In the following, we present this new system

with an emphasis on the major differences compared to the previous version of the instrument used at the University of Bern.

Our new measurement system combines a custom-made extraction unit with a gas chromatograph (GC) equipped with a thermal conductivity detector (air), a flame-ionization detector ($CH_4$) and an electron capture detector ($N_2O$). The system is optimized for the measurement of small amount of analytes, enabling to use as little as ∼20 g of EDC ice core samples. The major modifications compared to the previous version of the instrument are the change from a melt-refreeze extraction method

(e.g. Flückiger et al., 1999, 2002, 2004; Spahni et al., 2005; Schilt et al., 2010a, b; Baumgartner et al., 2012; Schilt et al., 2013; Baumgartner et al., 2014; Loulergue et al., 2008) to a continuous extraction under vacuum, according to the procedure described in Schmitt et al. (2014), as well as the complete renewal of the standardization procedure.

In our extraction unit, air released from the ice during the melting phase (immersion in a water bath at ∼293 K) is continuously adsorbed on an activated charcoal trap held at 77 K using liquid nitrogen. This extraction technique ensures low $CH_4$

and $N_2O$ partial pressures above the meltwater, minimizing dissolution during the melting, thereby avoiding the need for a refreezing step. To calibrate our measurements, we use a set of three standard gases provided by the National Oceanic and



Atmospheric Administration, covering the typical glacial-interglacial concentration range for $CH_4$ (358.88 ± 0.16 ppb, 838.59 ± 0.28 ppb, and 1729.30 ± 0.34 ppb) and $N_2O$ (187.10 ± 0.12 ppb, 194.13 ± 0.12 ppb, and 300.20 ± 0.12 ppb). The standards are referenced to the World Meteorological Organisation mole fraction scales: WMOX2004A scale ($CH_4$) and NOAA-2006A

($N_2O$) (Dlugokencky et al., 2005; Hall et al., 2007).

To increase the sample throughput, standard gases are injected directly into the GC system, bypassing the extraction line employed for ice core samples. We periodically inject standards over gas-free ice samples to account for contamination along the extraction line, determined as the mean offset between the two injection pathways (line offset). For $CH_4$, the line offset depends linearly on concentrations ($R^2$ = 0.99) leading to a downward revision of our measured values by up to 5 ppb (for

concentrations ranging between 350 and 700 pbb). For $N_2O$, the line offset is constant and leads to a downward revision of the measured values by 4 ppb.

The construction of composite records is complicated by offsets between the datasets, where our results appear 29 ± 7 ppb ($CH_4$) and 18 ± 2 ppb ($N_2O$) higher than previous data. The offset is calculated as the mean of the residuals between splines with cut-off periods of 10 thousand years fitted through the datasets. We computed splines according to Enting (1987),

using the same routine as Beck et al. (2018), where each spline is the average of 1000 iterations with data points varied within a normal distribution inside their $1\sigma$ uncertainty range. Because we are unable to experimentally compare the accuracy of different instruments using different extraction procedures, we adopt an *ad hoc* approach and subtract 29 ± 7 ppb and 18 ± 2 ppb from our new $CH_4$ and $N_2O$ results, respectively, to reach consistency with the datasets of Loulergue et al. (2008) and Schilt et al. (2010a). For the same reason, we do not correct for gravitational and thermal fractionation in the firn.

The precision of our new data amounts to 10 ppb for $CH_4$ and 6 ppb for $N_2O$ and is calculated as the square-root of the sum in quadrature of the individual uncertainties associated with the analytical procedure, the line offset correction and the inter-dataset correction. The analytical part corresponds to the standard deviation ($1\sigma$) of standards injected over gas-free ice samples. The uncertainty of the line offset correction corresponds to the standard deviation ($1\sigma$) of the mean offset between the two injection pathways (as described above). The uncertainty of the inter-dataset correction is the standard deviation ($1\sigma$) of

the mean residuals between the splines. For the existing EDC data and the isotopic measurements, the precision amounts to 10 ppb ($CH_4$), 4 ppb ($N_2O$), 0.22 ‰ ($\delta^{15}N(N_2O)$), and 0.34 ‰ ($\delta^{18}O(N_2O)$) (Loulergue et al., 2008; Schilt et al., 2010a; Schmitt et al., 2014).

## 3   Results

Our centennial-scale records show the progressions of the overall ∼370 ppb and ∼60 ppb increase in $CH_4$ and $N_2O$ concen-

tration, respectively, over the penultimate deglaciation (Fig. 1). Our $CH_4$ data identifies an outlier at 139.9 ka BP, in the dataset by Loulergue et al. (2008), ∼80 ppb higher than adjacent samples. The $N_2O$ data appear substantially more scattered in the early part of the record, especially around the penultimate glacial maximum (∼145–140 ka BP), where several high-amplitude spikes reach up to ∼300 ppb. This scatter tends to decrease with younger ages. The spikes appear in the interval where the isotopic composition of $N_2O$ becomes enriched in $\delta^{15}N(N_2O)$ (up to 21 ‰) and depleted in $\delta^{18}O(N_2O)$ (down to 41 ‰),





compared to the relatively steady values in the range of ~11–14 ‰ and ~45–47 ‰, respectively, observed for 125–136 ka BP
(Fig. 1).

Our records display several fluctuations standing out in the overall evolution of the $CH_4$ and $N_2O$ concentrations (Fig. 1).
For $CH_4$, we distinguish between periods of relatively gradual increase and periods marked by more abrupt fluctuations based
on mean rates of change observed in the ice core record over the corresponding time period. Abrupt $CH_4$ rises are identified at

~134 and ~128 ka BP. The 134-ka event exhibits an increase at a mean rate of 34 ppb per century, in the interval from ~134.0
to 133.8 ka BP, and a decrease at a mean rate of 23 ppb per century, in the interval from ~133.8 to 133.6 ka BP. During the
rising limb, concentrations increase from ~440 ppb to ~510 ppb before declining back to ~460 ppb. The 128-ka event consists
of a ~190 ppb increase (from ~530 to 720 ppb, about half of the deglacial change) proceeding in ~300 years (~128.9–128.6
ka BP). This feature exhibits the highest mean growth rate observed in our deglacial $CH_4$ record (mean of 61 ppb per century).

Beside the 134 and 128-ka events, $CH_4$ was also released relatively fast during the initial deglacial rise (~139.6–138.7 ka BP,
mean of 7 ppb per century) compared to the other time periods of more gradual increases (mean of 3 ppb per century from
~135.6 to 134.0 ka BP and from ~132.0 to 128.9 ka BP). However, the mean growth rate at this time is still well below those
of the abrupt fluctuations.

The evolution of $N_2O$ concentrations alternates between periods of plateaus and well-marked fluctuations (Fig. 1). Similar

to $CH_4$, a feature is resolved at ~134 ka BP, exhibiting the highest rate of change observed in our deglacial $N_2O$ record (mean
of 12 ppb per century during the rising limb). Concentrations increased in ~200 years (134–133.8 ka BP) from ~210 to 240
ppb before stabilizing for ~200 years and declining from ~133.6 ka BP onwards. The decay phase lasted for ~1000 year
(~133-6–132.6 ka BP) at a mean rate of 3 ppb per century. The 128-ka event is also imprinted in our $N_2O$ record (~129.0–
128.2 ka BP) and is characterized by rising concentrations from ~240 to ~270 ppb at a mean growth rate of 3 ppb per century.

In addition, an increase is identified at ~130.5 ka BP (~130.5–129.8 ka BP), where concentrations rose from ~220 to 245 ppb
at a mean growth rate of 3 ppb per century. The 130.5-ka and 128-ka events are separated by a plateau that lasted 800 years.

Overall, the improved resolution of our records allowed us to identify features hidden in the current $CH_4$ and $N_2O$ EDC
datasets. In particular, the 134-ka event and the $N_2O$ increase at ~130.5 ka BP are resolved for the first time. Retrieving $CH_4$
and $N_2O$ concentrations from the same samples enable us to study the relative phasing of both trace gases in the course of these

events without age uncertainty. At the onset of the 134 and 128-ka events, the rise in both trace gases occur simultaneously. In
contrast, the 130.5-ka event in the $N_2O$ record is not accompanied by a concomitant fluctuation in $CH_4$ concentrations.

## 4   Discussion

Interpreting our records in terms of atmospheric variability requires a closer inspection of the extreme values observed in the
$CH_4$ and $N_2O$ records. The $CH_4$ data point at 139.9 ka BP is measured in the section characterized by the widest GAD in

our record. Using the tentative approach of Nehrbass-Ahles et al. (2020) yields a mean width of the GAD estimated at ~220
years for the interval from 139 to 141 ka BP. The adjacent data points are 160 years older and 173 years younger than the
extreme measurement, i.e., the timescale of the hypothetic fluctuation at 139.9 ka BP is smaller than the GAD. This feature





is therefore unlikely to represent atmospheric variability and has to be considered as an outlier, likely resulting from the analytical procedure. Such signals may also result from layered bubble trapping (Rhodes et al., 2016; Fourteau et al., 2017, 2020). However, the outlier is measured in a period of otherwise stable concentrations, where neither early nor late pore closures are expected to generate an anomaly (Rhodes et al., 2016; Fourteau et al., 2017). Secondly, similarly high concentrations are not observed in our record before the 134-ka event. This would imply an age anomaly (between the layers enclosing gas of abnormal age and the layers enclosing gas of the corresponding age) that is unrealistically high compared to the characteristic age anomaly reported by Fourteau et al. (2017) for the Vostok ice core (~200 years). Accordingly, we regard this measurement as an analytical outlier at this point and exclude it from further analysis.

The large variability observed in the early part of the $N_2O$ record (Fig. 1) is unlikely to reflect atmospheric fluctuations given the atmospheric lifetime of $N_2O$ of $116 \pm 9$ years (Prather et al., 2015). Elevated concentrations and disproportionately high $N_2O$ variability have been observed in many instances for ice samples rich in dust in both Antarctic and Greenland ice cores and are attributed to in situ production (Flückiger et al., 1999; Sowers, 2001; Flückiger et al., 2004; Spahni et al., 2005; Schilt et al., 2010a, 2013; Fischer et al., 2019). Measurements in the Vostok ice core by Sowers (2001) demonstrated this excess $N_2O$ production to have a strong imprint on both $\delta^{15}N(N_2O)$ and $\delta^{18}O(N_2O)$. For some of our EDC samples we observe the same systematic deviations as for the Vostok samples of that period, with $\delta^{15}N(N_2O)$ and $\delta^{18}O(N_2O)$ values that are heavier and lighter, respectively, than the typical atmospheric value. The coupling with dust is the basis of an empirical artifact detection method, applicable to EDC samples, considering measurements for depth intervals where dust concentrations exceed an arbitrary threshold of 300 $\mu$g kg$^{-1}$ as affected by in situ production (Spahni et al., 2005; Schilt et al., 2010a). We follow this approach and define 134.5 ka BP, a slightly younger age than the last value considered as unbiased by Schilt et al. (2010a), as the boundary for the section affected by artifacts. In summary, we refrain from interpreting $N_2O$ data points older than 134.5 ka BP as reflecting atmospheric variability.

The younger part of our records are used to study the nature of the $CH_4$ and $N_2O$ fluctuations during TII and to compare with TI (Fig. 2). For such comparisons, we take the WAIS Divide ice core record (WD) (Rhodes et al., 2015) and the composite dataset of Fischer et al. (2019) as benchmarks for $CH_4$ and $N_2O$ concentrations, respectively, in the last deglaciation. The nature of the fluctuations is assessed by analyzing the amplitude and timescale of the individual features as well as the background climate in which they occur, inferred from complementary climate proxies (Fig. 3).

A feature common to TI and TII is the pronounced increase in $CH_4$ and $N_2O$ concentrations to interglacial levels at the end of the respective deglaciation. The 128-ka event marking the onset of the Last Interglacial (LIG) appears as an analogue of the rise at the end of the Younger Dryas (YD), marking the onset of the Holocene. $CH_4$ and $N_2O$ concentrations reached in the early interglacial times are approximately similar for the LIG and the Holocene (~720–740 ppb and ~270 ppb, respectively). The WD data show the onset of the $CH_4$ rise at the end of the YD to start from lower values (~480 ppb compared to ~530 ppb at ~128 ka BP) making this increase larger in magnitude by ~70 ppb. Moreover, the somewhat shorter timescale of this increase (less than 200 years compared to ~300 years at ~128 ka BP in our data) and the overshoot at the end of the event can, at least partly, be explained by the smaller extent of smoothing in the WD data compared to our records (Rhodes et al., 2015;





Nehrbass-Ahles et al., 2020). For $N_2O$, the amplitude and duration of the two events are approximately similar, exhibiting a ~30 ppb rise (from ~240 to 270 ppb) in ~800–900 years.

At the end of the YD and at ~128 ka BP, $CH_4$ and $N_2O$ concentrations rise in parallel with the main resumption of the
AMOC, indicated by the evolution of the isotopic ratio of neodymium 143 and 144 ($\varepsilon_{Nd}$) (Deaney et al., 2017; Böhm et al., 2015), and the associated northward shift of the ICTZ indicated by the evolution of the isotopic composition of speleothem calcite ($\delta^{18}O(CaCO_3)$) (Cheng et al., 2009, 2016) (Fig. 4), in a manner consistent with DO-like variability. The abrupt $CH_4$ rise reflects the response of terrestrial emissions likely from NH tropical wetlands. The simultaneity of the $N_2O$ increase indicates that terrestrial emissions contributed, at least partly, to the 128-ka event, similar to what has been demonstrated for TI on the
basis of an isotopic deconvolution (Fischer et al., 2019). The data of Deaney et al. (2017) suggest the main AMOC resumption to have occurred at this time. Therefore, it can be assumed that marine sources also played a role, consistent with what has been shown for the end of the YD (Schilt et al., 2014; Fischer et al., 2019). Higher resolution measurements of the isotopic composition of $N_2O$ combined with a deconvolution, similar to Schilt et al. (2014) and Fischer et al. (2019), are needed to quantitatively determine the relative contribution of the sources during the 128-ka event.

The 128-ka event is preceded by a phase of rising $N_2O$ concentrations in the interval from 130.5 to 129.8 ka BP at the end of HS11 (135–130 ka BP) (Marino et al., 2015) (Fig. 1). This is reminiscent of the pattern of late stadial increases, where $N_2O$ concentrations rose before DO-like Greenland temperature and $CH_4$ changes (Schilt et al., 2013). The timescale of the 130.5-ka event is in the range of the duration typically observed for these episodes (~0.5–2 millennia) (Schilt et al., 2013). On the other hand, the $N_2O$ growth rate during our event appears slightly larger (~3 ppb compared to ~1 ppb per century).

The evolution of $N_2O$ concentrations from 130.5 ka BP to the onset of the 128-ka event is remarkably coeval, within dating uncertainty, with the variability of $\delta^{18}O(CaCO_3)$ and of the isotopic composition of atmospheric oxygen ($\delta^{18}O(O_2)$) (Landais et al., 2013; Cheng et al., 2009) (Fig. 4), also displaying an initial period of change followed by plateaus. At this timescale, $\delta^{18}O(CaCO_3)$ and $\delta^{18}O(O_2)$ reflect changes in the low latitude hydrological cycle driven by shifts in the ICTZ (Landais et al., 2010; Cheng et al., 2009). Taken at face value, they indicate a small intensification of the tropical hydroclimate at ~130.5
ka BP prior to the DO-like fluctuation at ~128 ka BP. This intensification has been interpreted as a transition from a HS to a DO stadial (Landais et al., 2013) and might be a consequence of the partial AMOC resumption reported in Böhm et al. (2015). In fact, the isotopic ratios of protactinium 231 and thorium 230 ($^{231}Pa$ / $^{230}Th$), reflecting the overturning rate of the AMOC, shows a decline to interglacial values substantially earlier than the decrease in $\varepsilon_{Nd}$ occurring around ~128 ka BP (Böhm et al., 2015) (Fig. 3). The decoupling of the two oceanic tracers during part of TII has been interpreted as a change
in the AMOC regime from the *off* mode, characteristic of HS (suppressed convection of northern-sourced water, extremely reduced overturning rates, high $\varepsilon_{Nd}$, and high $^{231}Pa$ / $^{230}Th$) to the *cold* mode (shallow convection, vigorous overturning rate, high $\varepsilon_{Nd}$, and low $^{231}Pa$ / $^{230}Th$) (Böhm et al., 2015), prevailing around glacial maxima. Additional evidence arguing for a partial resumption of the AMOC is given by the evolution of the isotopic composition of atmospheric nitrogen ($\delta^{15}N(N_2)$), a proxy for Antarctic temperature, displaying a marked leveling off at ~130.5 ka BP, consistent with a SH temperature response
to a slightly enhanced northern heat advection by the AMOC (Landais et al., 2013; WAIS Divide Project Members, 2015; EPICA Community Members, 2006; Stocker and Johnsen, 2003; Pedro et al., 2018; Buizert et al., 2018) (Fig. 4).



Taking the evidences together, we propose that the 130.5-ka event constitutes a response to the transition from a HS to a DO stadial, where the rise in concentrations is associated with the partial AMOC resumption. The lack of a concomitant $CH_4$ fluctuation suggests that only changes in marine sources contributed to the $N_2O$ increase. Overall, the 130.5-ka event fits into

the framework of the late stadial events (Schilt et al., 2013, 2014; Fischer et al., 2019) and can be viewed as an analogue of the late HS1 rise during TI. However, the $N_2O$ emission rate was slightly larger than evidenced for HS1 (as well as for similar events during the last glacial period). The attribution of the 130.5-ka feature to the pattern of late stadial increases would be strengthened by additional measurements of $\delta^{15}N(N_2O)$ and $\delta^{18}O(N_2O)$, allowing the unambiguous identification of the dominant source contributing to this event.

Turning to the 134-ka event, its occurrence within HS11 and the properties of the $CH_4$ increase (duration and amplitude) are reminiscent of the intra-stadial pattern of variability evidenced by Rhodes et al. (2015). Intra-stadial fluctuations resolved in the WD ice core fully developed within ∼200–300 years (Rhodes et al., 2015). The broader GAD of the EDC ice core (∼170 years for our record) implies that any such intra-stadial features have to appear strongly dampened in our data. This is supported by continuous $CH_4$ measurements in the Vostok ice core, demonstrating the absence of the characteristic overshoot resolved in

the WD ice core for the HS4 intra-stadial fluctuation (Fourteau et al., 2020; Rhodes et al., 2015). Therefore, the sharpness and amplitude of the 134-ka event is not consistent with the picture of a substantially smoothed version of an intra-stadial fluctuation. The ∼70 ppb rise would translate into a WD signal exceeding by far the amplitude range (∼30–55 ppb) of the HS1, HS2, HS4 and HS5 fluctuations (Rhodes et al., 2015). Secondly, intra-stadial $CH_4$ variability is also characterized by abrupt $CO_2$ jumps, millennial-scale increase in $\delta^{18}O(O_2)$ and the absence of concomitant $N_2O$ variability (Bauska et al., 2016,

2018; Marcott et al., 2014; Fischer et al., 2019; Schilt et al., 2013, 2010a; Guillevic et al., 2014). The simultaneous occurrence of the $CH_4$ and $N_2O$ pulses at ∼134 ka BP and the lack of any fluctuation in the $\delta^{18}O(O_2)$ record (Landais et al., 2013) contradict these observations (Fig. 3 and 4). The 134-ka event is therefore likely to have resulted from different mechanisms than those driving intra-stadial variability and, consequently, cannot be considered as an analogue of the 16.15-ka event that developed within HS1 (Fig. 2).

On the other hand, simultaneous rises of atmospheric $CH_4$ and $N_2O$ concentrations are observed during DO-like fluctuations. In the last glacial cycle, interstadials typically lasted for ∼1500 years with durations ranging from 240 to 14,380 years for Greenland Interstadial 3 and Greenland Interstadial 23, respectively (Rasmussen et al., 2014). The onset of these interstadials were accompanied by increases in concentration ranging from 50 to 260 ppb ($CH_4$) and from 10 to 40 ppb ($N_2O$) (Baumgartner et al., 2014; Schilt et al., 2013; Flückiger et al., 2004). The amplitude and duration of the $CH_4$ pulse at ∼134 ka BP (∼400

years, ∼70 ppb) are well comparable to those observed for DO3, DO6, DO9, and DO10, where amplitudes are smaller or equal to 100 ppb (Baumgartner et al., 2014) and durations are shorter than 700 years (Rasmussen et al., 2014). For $N_2O$, the amplitude of the 134-ka event compares well with those of DO-like fluctuations (Flückiger et al., 2004).

  Associating our event to a DO-like pattern of variability requires evidence for a northward shift of the ICTZ and reinvigoration of the AMOC. Synchronous with the 134-ka event, within dating uncertainty, we observe a short-lived negative excursion

in speleothem $\delta^{18}O(CaCO_3)$ records (Cheng et al., 2006) (Fig. 4) as well as fluctuations in proxies reflecting salinity and runoff intensity in the Bay of Bengal (Nilsson-Kerr et al., 2019). These data indicate a transient strengthening of the tropical monsoon





systems consistent with a northward shift of the ICTZ. Concerning the behavior of the AMOC, we are not aware of studies elaborating on a possible reinvigoration at this time. Nevertheless, oceanic tracers exhibit a small and short-lived fluctuation in the time interval 133–132 ka BP (on the timescale of Böhm et al. (2015)) before reaching their maximum HS11 values (Böhm

et al., 2015). On the updated chronology of the sediment core ODP Site 1063 (Deaney et al., 2017), the negative excursion in the $\varepsilon_{Nd}$ record of Böhm et al. (2015) coincides with a comparably small value in the data of Deaney et al. (2017) (Fig. 3). However, the revised chronology places the excursion in $\varepsilon_{Nd}$ and $^{231}Pa$ / $^{230}Th$ substantially earlier (onset at ∼137.4 ka BP, corresponding to a shift by ∼4.9 thousand of years with respect to the timescale of Böhm et al. (2015)) than the 134-ka event (Fig. 3 and 4). Since the timescale of the sediment core is tuned to the Antarctic Ice Core Chronology (AICC2012) (Bazin

et al., 2013; Veres et al., 2013) within 400 years, the fluctuations resolved in the ice and marine cores shall, in principle, not be considered as synchronous. However, the alignment with the AICC2012 is performed using only one tie point between $CH_4$ and the isotopic composition of planktonic foraminifera (at ∼128.7 ka BP, corresponding to the abrupt $CH_4$ increase into the LIG) (Deaney et al., 2017). Therefore, we keep the possibility open that the two timescales are less tightly aligned away from the tie point and suggest that an occurrence of the excursions in oceanic proxies synchronous with the fluctuations of

$\delta^{18}O(CaCO_3)$ and trace gases might still be possible. DO-like variability is typically also imprinted in the $\delta^{15}N(N_2)$ record from Antarctic ice cores as well as in the $\delta^{18}O(O_2)$ record. However, changes in these proxies are generally subtle compared to those of $CH_4$ and $\delta^{18}O(CaCO_3)$, especially for $\delta^{18}O(O_2)$ (Landais et al., 2010). Accordingly, the duration or amplitude of the 134-ka event might not have been sufficient to produce a discernible signal in the records of Landais et al. (2013) (Fig. 3). Taking the proxy evidences together, we speculate that a brief and small-scale resumption of the AMOC might have occurred

within HS11, accounting for the northward ICTZ shift and the rise in atmospheric $CH_4$ and $N_2O$ concentrations, consistent with the picture of DO-like variability. We acknowledge that our interpretation is limited by the relatively coarse resolution of $\varepsilon_{Nd}$ and $^{231}Pa$ / $^{230}Th$ as well as by the uncertainty arising from cross-dating sediment and ice core records. It should finally be noted that the simultaneity of the $CH_4$ and $N_2O$ increase and the duration of the rising phase is consistent only with a contribution from terrestrial $N_2O$ sources (Fischer et al., 2019).

Should our interpretation hold, the 134-ka event can be considered as a short DO-event, contrasting with the marked Bølling–Allerød (BA) fluctuation during TI (Fig. 2). This raises the question of why the oceanic circulation was unable to recover in the same way as during the BA. Because the 134-ka event coincides, within dating uncertainty, with the occurrence of Meltwater Pulse 2B (MWP–2B) (Fig. 3), it would be tempting to link the quenching of the emerging interstadial at ∼134 ka BP to freshwater forcing. MWP-2B constitutes the major melting event in the course of TII. As a matter of fact, it contributed ∼70

m of sea-level rise (∼70 % of the deglacial change) and mainly entered into the North-Atlantic along with substantial volumes of icebergs, as indicated by the elevated occurrence of ice rafted debris recorded in sediment cores at this time (Marino et al., 2015; Grant et al., 2014; Skinner and Shackleton, 2006) (Fig. 3).

However, the concept of freshwater forcing as a trigger of AMOC shutdown (either as iceberg discharges during HS or MWPs during glacial terminations) has become a matter of intense debate. Firstly, the history of MWPs is decoupled from that

of the AMOC and Greenland temperature in the course of TI (most notably exemplified by the absence of any such pulses for the YD stadial) (Tarasov and Peltier, 2005; Stanford et al., 2006). Secondly, iceberg discharge events within HS are consistently



lagging behind oceanic circulation and Greenland temperature changes (Barker et al., 2015; Henry et al., 2016). The current paradigm rather considers freshwater forcing as resulting from AMOC declines. This view is supported by modeling and experimental studies demonstrating that the interior of the ocean accumulates heat at times the AMOC collapses (Galbraith

et al., 2016; Pedro et al., 2018; Bereiter et al., 2018; Baggenstos et al., 2019), constituting a potent forcing for destabilizing glacial ice sheets and producing bursts of meltwater (Flückiger et al., 2006; Marcott et al., 2011; Clark et al., 2020; Galbraith et al., 2016). In summary, we are currently unable to propose a mechanism accounting for the relative brevity of the 134-ka event.

Summarizing the evidences, we note that the outstanding fluctuations in $CH_4$ and $N_2O$ concentrations during the penultimate

deglaciation are all instances of recurrent modes of variability, also evidenced during the last deglaciation as well as during the last glacial period.

## 5  Conclusions

In the present study, we increased the resolution of the deglacial $CH_4$ and $N_2O$ records, allowing us to derive composite datasets covering TII (140–128 ka BP) at average resolutions of $\sim$100 years. Our results display fluctuations standing out of

the overall transition of $CH_4$ and $N_2O$ concentrations to interglacial conditions. The most prominent one is resolved at $\sim$128 ka BP and constitutes an analogue of the rise at the end of the YD during the last termination. We assume that terrestrial and marine sources contributed to the $N_2O$ increase at this time. Additionally, we now entirely identify the 134 and 130.5-ka event. We link the latter to the pattern of late stadial $N_2O$ increase, where changes in marine emissions are likely to be the only contributor. The former is regarded as a short DO-like fluctuation, whose timescale indicates that only terrestrial $N_2O$ sources

likely contributed to the increase.

*Data availability.* Data will be made available online on the NOAA paleoclimate and PANGAEA databases.

*Author contributions.* The present study was designed by T.F.S, H.F and L.S. L.S and J.H performed the methane and nitrous oxide measurements. J.S provided the isotopic data. L.S wrote the text with inputs from all authors.

*Competing interests.* The authors declare that they have no conflict of interest.

*Acknowledgements.* The authors would like to thank Barbara Seth for the measurements of the isotopic composition of $N_2O$, Gregory Teste for assistance in cutting ice samples, as well as Michael Bock and Jan Strähl for the construction of the new $CH_4$ and $N_2O$ measurement system. We acknowledge financial support by the Swiss National Science Foundation (SNF project numbers 200020_172745 and





200020_172506). This work is a contribution to the *European Project for Ice Coring in Antarctica* (EPICA), a joint European Science Foundation/European Commission scientific program, funded by the European Union and by national contributions from Belgium, Denmark, France, Germany, Italy, The Netherlands, Norway, Sweden, Switzerland, and the United Kingdom. The main logistic support was provided by IPEV and PNRA. This is EPICA publication no. XX.



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





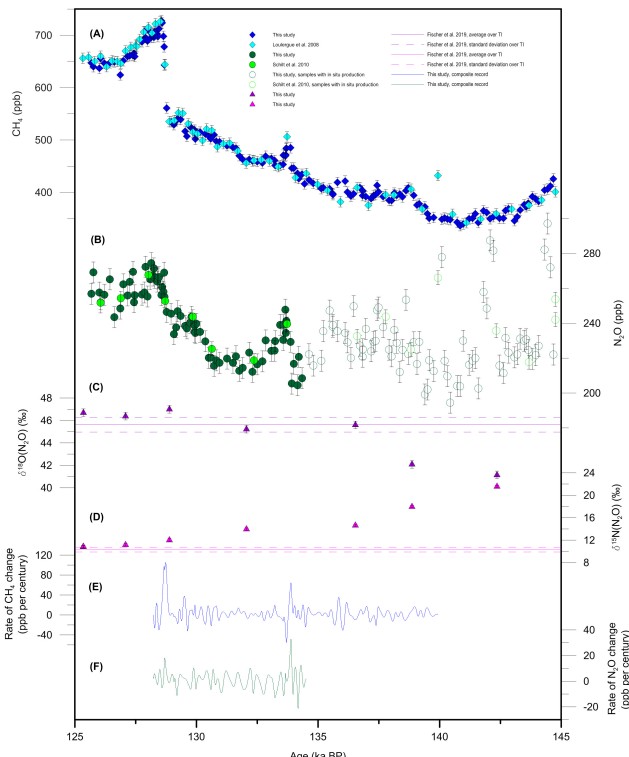

**Figure 1.** CH$_4$, N$_2$O, $\delta^{18}$O(N$_2$O) and $\delta^{15}$N(N$_2$O) records from the EDC ice core on the AICC2012 timescale (Bazin et al., 2013). The vertical bars represent the uncertainty of the measurements. **(A):** Composite CH$_4$ record with published data (light blue, Loulergue et al. (2008)) and new measurements (this study, dark blue) after offset correction. **(B):** Composite N$_2$O record with published data (light green, Schilt et al. (2010a)) and new measurements (this study, dark green) after offset correction. The empty symbols illustrate the data points considered as affected by in situ production. **(C):** $\delta^{18}$O(N$_2$O) record. For comparison, a solid and a dashed lines are included, representing the average and standard deviation (1$\sigma$), respectively, of the isotopic values during TI. **(D):** $\delta^{15}$N(N$_2$O) record, including solid and dashed lines as in **(C)**. **(E):** Rates of CH$_4$ change, calculated by differentiating splines (Enting, 1987) (cut-off period = 200 years) fitted through the composite record. **(F):** Rates of N$_2$O change, for the section unbiased by artifacts, calculated as in **(E)**.

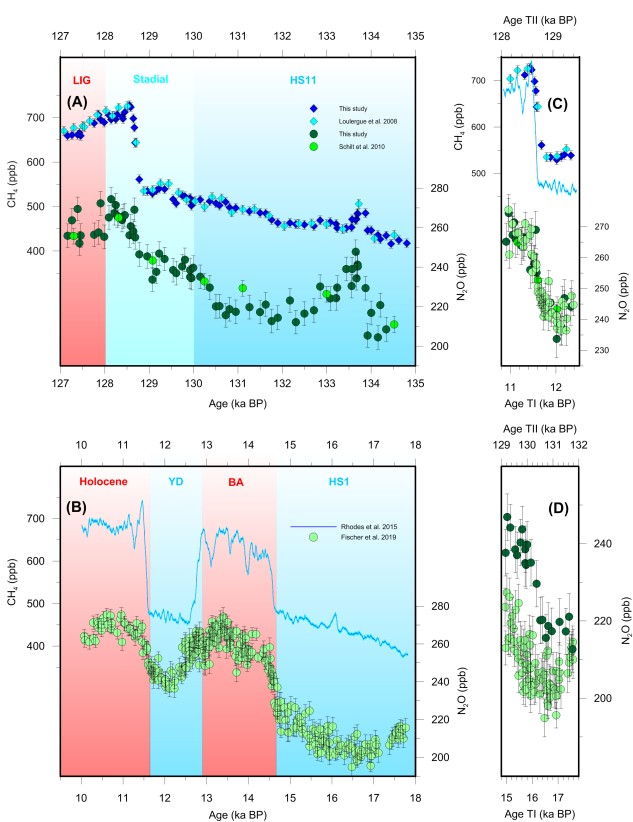

**Figure 2.** Evolution of the $CH_4$ and $N_2O$ concentrations during TII and TI. The left panels show the overviews for the individual deglaciation and the right panels show in details the fluctuations compared in this study. The shadings delineate the climatic events mentioned in the text and distinguish between time of collapsed (darker blue), reduced (paler blue) and vigorous AMOC (red). **(A):** Composite $CH_4$ (top) and $N_2O$ (bottom) records for TII as in Fig.1. **(B):** WD continuous $CH_4$ record (top) (Rhodes et al., 2015), on the WAIS Divide deep ice core chronology (Buizert et al., 2015; Sigl et al., 2016), and composite $N_2O$ record (bottom) (Fischer et al., 2019), on the AICC2012 timescale (Veres et al., 2013), for TI. **(C):** Superposition of the 128-ka event with the fluctuation at the end of the YD for $CH_4$ (top) and $N_2O$ (bottom) (symbols as in **(A)** and **(B)**). **(D):** Superposition of the late HS11 and HS1 $N_2O$ rises (symbols as in **(A)** and **(B)**).





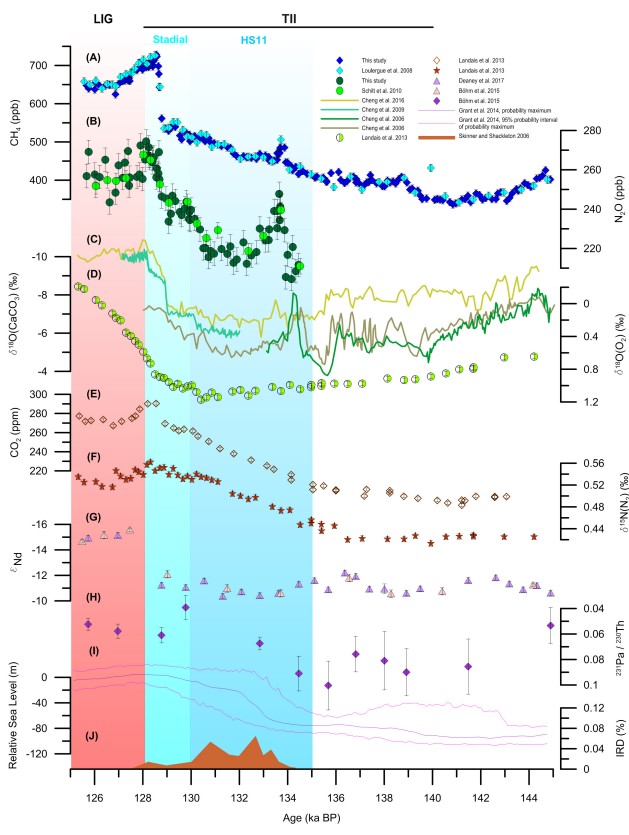

**Figure 3.** Evolution of the $CH_4$ and $N_2O$ concentrations as well as this of complementary climate proxies during TII. The shadings delineate the climatic events as described in Fig. 2. The individual records are plotted on their own timescales, with the vertical bars representing the uncertainty of the measurements as reported in the references. **(A)**: Composite $CH_4$ record (this study, as in Fig.1). **(B)**: Composite $N_2O$ record (this study, as in Fig.1). **(C)**: Speleothems $\delta^{18}O(CaCO_3)$ records: Sanbao SB25 (light green) (Cheng et al., 2009), Hulu Cave MSX (khaki) (Cheng et al., 2006), Hulu cave MSP (dark green) (Cheng et al., 2006), and Sanbao-Dongge composite (pale yellow) (Cheng et al., 2016). Each speleothem record is plotted on its individual radiometric timescale. **(D)**: $\delta^{18}O(O_2)$ from the EDC ice core on the AICC2012 timescale (Landais et al., 2013). **(E)**: $CO_2$ from the EDC ice core on the AICC2012 timescale (Landais et al., 2013). **(F)**: $\delta^{15}N(N_2O)$ from the EDC ice core on the AICC2012 timescale (Landais et al., 2013). **(G)**: Composite $\varepsilon_{Nd}$ from the sediment core ODP Site 1063 (Deaney et al., 2017), including data points from Böhm et al. (2015) (purple) and new measurements from Deaney et al. (2017) (pale pink), on the timescale of Deaney et al. (2017). **(H)**: $^{231}Pa$ / $^{230}Th$ record from the sediment core ODP Site 1063 (Böhm et al., 2015), on the timescale of Deaney et al. (2017). **(I)**: Relative sea level stand from the Red Sea synchronized on a radiometric timescale, including the maximum probability curve (purple) and its 95% confidence interval (magenta) (Grant et al., 2014). **(J)**: Ice-rafted debris (IRD) from Skinner and Shackleton (2006) on the radiometric timescale of Marino et al. (2015).



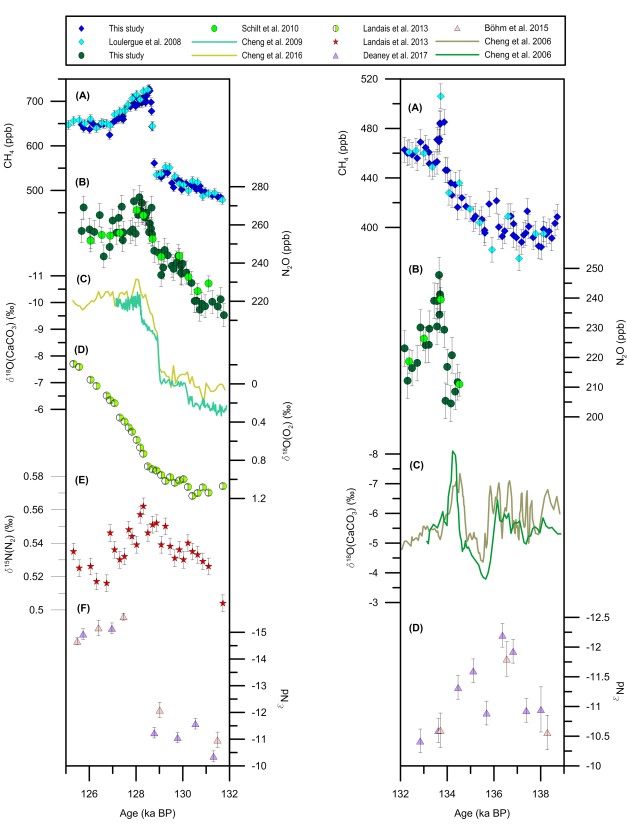

**Figure 4.** Details of the 128 (left panel), 130.5 (left panel) and 134-kyr events (right panel) for CH₄ **(A)**, N₂O **(B)**, and key climate proxies (from **(C)** onwards), symbols as described in Fig. 3.