# Peer review of "$CH_4$ and $N_2O$ fluctuations during the penultimate deglaciation"

_Climate of the Past, 2020_

## Referee Comment (RC1) · Anonymous Referee #1 · 19 Nov 2020

This manuscript describes a new protocol for the measurement of CH4 and N2O concentrations on small samples of ice. The new method is applied to samples from the EDC ice core and enables new composite records to be developed covering the penultimate deglacial period (Termination 2).

I'm sorry to say that I was rather underwhelmed by this paper. I had expected to see exciting new records with new insights and an opportunity to learn more about T2. Indeed, there are a couple of novel features reported but nothing very exciting. In all I suspect this paper will add to a growing body of studies dealing with T2 that will ultimately (but not yet) lead to an increase in understanding.

The authors describe a new analytical approach, an important step forward which deserves to be documented. However, the authors point to a significant offset with mea-

surements made by earlier methods and end up correcting their new data in an ad hoc fashion (minimising the difference between the various datasets). This operation implies that the authors have little confidence in the absolute values of their results, and this is obviously alarming. Is there really no way to produce a standard that can be used to cross calibrate between techniques?

The data themselves are interesting but offer little new insight beyond increasing temporal resolution of older records from the same ice core. The new composite record of N2O does improve on the older record but the CH4 record merely confirms that details which previously were suspected are actually real.

Some confusing nomenclature is developed here that leads to ambiguity and a loss of logic. For example, the authors distinguish 'late stadial' from 'intra-stadial' variability, which is fine on the face of it but becomes confused when they discuss the 134ka event, which occurs within a stadial event (HS11) but is apparently not an 'intra-stadial' event (somewhat of an oxymoron?). This confusion comes from the fact that the authors are using a non-specific term (intra-stadial) to define a specific mode of variability that was described in a paper(s) by Rhodes et al. (2015) and was previously argued to be related to strengthening of southern hemisphere monsoon systems and a southward shift of the ITCZ (as opposed to a northward shift, which might be expected with the abrupt transitions from stadial to interstadial state). Perhaps the authors need to find an alternative (more descriptive) name for these 'type' of event.

More confusion occurs with the discussion of the 130.5ka 'event', which the authors suggest might represent a transition from an HS event to a D-O stadial. This description doesn't make much sense to me I'm afraid. A Heinrich-stadial (HS) has been defined as a stadial that contains a Heinrich event (HE). Thus the label HS11 implies that this cold interval contains the Heinrich event HE11. It makes no sense to imply that HS11 can change to a regular stadial (that does not contain an HE) once Heinrich Event 11 has ended (if that is what happens). I suppose you could argue that HE11 (Heinrich Event 11) ended before the end of HS11 but HS11 does not become a regular stadial

once the HE is over. I realise that the present authors took this idea from an earlier paper by Landais et al., (2013) but it makes no more sense in that paper.

Perhaps I am missing something obvious but the authors' discussion and figures describe an event at 128ka which I believe occurred between 128.6 to 128.7ka, in which case they should perhaps round up to 129ka rather than down to 128ka. Rounding to the nearest kyr and adding a tilde would be fine in the text ($\sim$129ka) but this is not something to do in a figure. In figures 2 and 3, the authors indicate a transition from stadial to interglacial conditions at 128ka that does not align with the rapid increase in CH4 that would more commonly be interpreted to indicate the end of a stadial period (cf end YD as shown in their Fig. 2).

My final major gripe concerns the authors' discussion and interpretation of their results. Most of the interpretation follows previous studies (e.g. rapid CH4 rise coincides with AMOC recovery at end of a stadial period) but where there is something new to discuss (e.g. the mysterious 134ka event) the authors provide a very slim argument for a major conclusion within their abstract (that this event was analogous to a D-O warming of MIS 3).

Below I outline some more comments in order of their appearance in the ms.

Abstract Line 5/6: "These features occurred in concert with reinvigorations of the Atlantic Meridional Overturning Circulation (AMOC) and northward shifts of the Intertropical Convergence Zone." Implies that the authors have identified at least 2 instances of this feature, which turns out to be a single instance (which was previously known about) and an additional instance that is substantially unsupported.

Line 8: (editorial) The authors' use of ka and ka BP needs attention; ka is fine on its own as shorthand for kyr ago or kyr BP (avoid ka ago or ka BP).

Line 8 and throughout: "…are assimilated to…". The meaning here is not clear – please check use of English throughout.

Main text Line 45: 'late stadial' – perhaps better as 'late HS'?

Line 46: "This mode of variability has been evidenced for the HS during the last glacial period..." Please check English.

Line 66-75: Much of this intro section feels more like discussion.

Results section: Much of the text here is unnecessary and feels verbose; you don't need to describe every inflection and quantify every rate - you have figures for this!

Line 129: "Abrupt CH4 rises are identified at ~134 and ~128 ka BP" Again, if rounding is to be used then rounding up from 128.6/7 to 129 ka would be more conventional.

Discussion Line 154: But the CH4 point at 139.9ka is from a previous study yes? Need to make this clearer.

Line 210: "...remarkably coeval...". Nothing remarkable here, perhaps approximately coeval.

Line 229: "Overall, the 130.5-ka event fits into the framework of the late stadial events (Schilt et al., 2013, 2014; Fischer et al., 2019) and can be viewed as an analogue of the late HS1 rise during TI." I am not convinced by this; the rise in N2O ~130.5ka is very similar in magnitude and rate as that ~129ka and yet the 'late stadial' rises documented for T1 and MIS 3 (Schilt et al., 2013) are significantly slower than the rather abrupt rise that accompanies the transition to interstadial conditions. Given that the 130.5ka rise is also separated from the 'end-stadial' rise by ~1000 years, which is also not noted for events during T1 or MIS 3 I think there is room for questioning the validity of this analogue.

Lines 235-284: I agree that something interesting happened ~134ka and that the CH4 and N2O evidence presented here, together with the speleo records and monsoon reconstruction reported by Nilsson-Kerr et al. (2019), suggest that something atmospheric was involved. The logical next step is to invoke a change in AMOC but there is no firm evidence for this. The authors argue (reasonably) that the age models employed for ODP 1063 by Bohm et al. (2015) and Deaney et al. (2017) are uncertain enough to allow significant wiggle room but the change in eNd that would involve (up to 2 epsilon units) is nearly half of that associated with the hypothesised AMOC resumption ~129ka. If eNd is taken at face value as a deep-water circulation tracer (which is not entirely free of problems) then why do we not find more evidence of such a large event ~134ka throughout the North Atlantic? Furthermore, the authors state that the timescale of CH4/N2O change across this event 'precludes' an oceanic source – so why call on an oceanic mechanism? Perhaps the authors need to soften their argument that the 134ka event is really such a good analogue for a D-O event (see earlier comment on Lines 5/6 of abstract).

The figures and fonts on these are painfully small.

References:

Böhm, E., Lippold, J., Gutjahr, M., Frank, M., Blaser, P., Antz, B., Fohlmeister, J., Frank, N., Andersen, M. B., and Deininger, M.: Strong and deep Atlantic meridional overturning circulation during the last glacial cycle, Nature, 517, 73–76, https://doi.org/10.1038/nature14059, 2015.

Deaney, E. L., Barker, S., and Van de Flierdt, T.: Timing and nature of AMOC recovery across Termination II and magnitude of deglacial CO2 change, Nat. Commun., 8, 1–10, https://doi.org/10.1038/ncomms14595, 2017.

Landais, A., Dreyfus, G., Capron, E., Jouzel, J., Masson-Delmotte, V., Roche, D. M., Prié, F., Caillon, N., Chappellaz, J., Leuenberger, M., Lourantou, A., Parrenin, F., Raynaud, D., and Teste, G.: Two-phase change in CO2, Antarctic temperature and global climate during Termination II, Nat. Geosci., 6, 1062–1065, https://doi.org/10.1038/ngeo1985, 2013.

Nilsson-Kerr, K., Anand, P., Sexton, P. F., Leng, M. J., Misra, S., Clemens, S. C., and Hammond, S. J.: Role of Asian summer monsoon subsystems in

the inter-hemispheric progression of deglaciation, Nat. Geosci., 12, 290–295, https://doi.org/10.1038/s41561-019-0319-5, 2019.

Rhodes, R. H., Brook, E. J., Chiang, J. C. H., Blunier, T., Maselli, O. J., McConnell, J. R., Romanini, D., and Severinghaus, J. P.: Enhanced tropical methane production in response to iceberg discharge in the North Atlantic, Science, 348, 1016–1019, https://doi.org/10.1126/science.1262005, 2015.

Schilt, A., Baumgartner, M., Eicher, O., Chappellaz, J., Schwander, J., Fischer, H., and Stocker, T. F.: The response of atmospheric nitrous oxide to climate variations during the last glacial period, Geophys. Res. Lett., 40, 1888–1893, https://doi.org/10.1002/grl.50380, 2013.

---

## Referee Comment (RC2) · Anonymous Referee #2 · 16 Dec 2020

Schmidely et al. provide new high-resolution CH4 and N2O datasets over the penultimate deglaciation from the EDC ice core, and based on their data suggest an interpretation of the last deglaciation that has more structure and richness that the "canonical" interpretation of just a long AMOC shutdown over H11 from 136 to 128 ka BP (Cheng et al., 2009; Clark et al., 2020). While the interpretation is somewhat speculative at times, I for one believe that they are mostly correct. It is always exciting to see that the climate is more complicated than we thought at first – it always pays off to evaluate ice core records at the highest possible resolution. I think the manuscript is suitable for publication after addressing some issues. In particular, the new analytical setup (a major achievement by itself) deserves some more attention, as well as the observed (and surprisingly large) offset with existing records.

[Figure]

(1) The 134 ka DO event

I am most excited about the discovery of a minor DO event early in the deglacial sequence (134 ka BP). When first looking at their data, I assumed the 134 ka CH4 features was the TII equivalent of the 16.2 ka Heinrich 1 CH4 feature identified by Rhodes et al. (2015). However, based on the Chinese speleothem record, as well as the size of the feature, Schmidely et al. convincingly argue that it is in fact a small DO event.

Personally, I think the authors should present this 134ka feature as an analog of DO2 – which likewise has an extremely short duration, and a (very!) small CH4 peak. The two have a similar orbital configuration, with very small NH summer insolation.

The small magnitude of the CH4 peak seems in line with the small NH insolation at 134 ka. (Baumgartner et al., 2014).

The authors struggle to explain the short duration of the event relative to the Bolling, but I think this is not necessarily hard. It has been shown (Buizert & Schmittner, 2015) that the duration of DO events scales strongly with Antarctic temperature (and following more recent work, presumably also with mean ocean temperature). The 134 ka DO event occurred much earlier in the TII deglacial sequence than the Bolling event in the TI sequence. Therefore, during the 134ka DO event global CO2 and Antarctic temperature were both lower than they were during the Bolling. These factors would predict a much shorter DO event at 134 ka than at 14.7 ka, as is indeed the observation of the authors.

The idea that meltwater "squelched" the DO event seems unlikely to be the complete explanation by itself, given that the Bolling was likewise accompanied by MWP-1A. Such a MWP is indeed perhaps the result of suddenly warming the NH via AMOC invigoration, thereby melting ice at the NH high-latitudes. However, Buizert and Schmittner (2015) show that under colder Southern Ocean background conditions (as during the 134 ka event), the AMOC is much more susceptible to meltwater perturbation. This may explain why the Bolling survived MWP-1A, and yet the 134ka event did not survive

MWP-2B.

(2) The analytical setup

The new setup is a major part of the paper, and has not been documented elsewhere. The line uses some new methodologies that have not previously been used for CH4/N2O analysis. It thus seems necessary to provide more details on this aspect of the paper. The authors should provide a schematic drawing of the line, and discuss its operation in more detail. For example, how is the sample calibration done, and how often? What is the precision of the new line from replicates? Do you calculate concentration using the air concentration from the thermal conductivity detector? What carrier gas is used? What is the "line offset"? How many samples can you analyze? Do you apply a solubility correction? Does the line produce total air content data?

Most importantly, the authors should perform a more thorough investigation of the large offset of the new setup with old data. The authors choose to correct the new data, implying they trust the old method better. What could cause such an offset? It would need to be understood if the setup is to be applied to periods of time where we do not have a reliable CH4 record already that can be used for offset correction – during such periods we need to be able to rely on the internal calibration of the line.

Perhaps all this material could be place in an appendix as to not affect the flow of the paper. This would be a valuable place of reference for future work that uses this setup.

(3) Line-by-line comments:

L8 and L9: What does "assimilate" mean in this context? Please clarify

L13: why compare it to the Bolling? Why not DO2 – based on background climate (insolation, CO2, mean ocean temp, etc) that may be a closer analog. Compared to DO2 it looks quite normal.

L34: What is the width of the gas age distribution using the recent estimate of Epifanio et al. (2020) who related it to Delta-age?

Page 2: The distinction between millennial and centennial CH4 events is somewhat arbitrary and perhaps even incorrect. Several of the DO events last only a few hundred years, and some of the H-event CH4 changes persist > 1000 years (like H4 and H5). Maybe just call them DO and H-event CH4 features? That would be much clearer.

L61: "believed to" should be "hypothesized to"

L73 is "though to be" driven exclusively by . . .

L78: maybe give a one-sentence description of the IRMS setup for N2O istopes

L91: acknowledge this is also the approach used by the Japanese lab at NIPR (Oyabu et al., 2020)

Section 2: More details are needed here, as outlined above.

L102: This is a very large offset – surely much greater than the specified precision of either line. Can you clarify? The approach is quite cavalier. An explanation of the offset is needed. Is this due to the new method? That is implied by the correction approach. The comparison in Fig. 1 implies that the offset may be concentration-dependent (both CH4 and N2O). Can you show a scatter plot of old vs. new (found with the spline method), with a linear fit? It appears it may not be a constant offset as applied.

L147: hidden in the previously published?

L152: could you add subheadings to the discussion? This is just one long block of text now. Maybe separate out the discussion of the 130.5 ka event and the 134 ka event, for example. There may be other subheadings that can be added. This would help structure the discussion better.

L155: what does "tentative approach" mean?

L155: How do you define the width? Please be specific. Some common metrics are the second moment of the distribution (or variance, or spectral width) or the FWHM (full width at half maximum). Just be specific how you define the width.

L185: the 128 ka may as well be an analogue of the Bolling transition that ended HS1.

L189: It looks like the 128ka event in CH4 is more similar to the Bolling transition in terms of magnitude and timing.

L190: I doubt the difference in magnitude is related to smoothing, since the step in CH4 lasts for millennia, so that both sites reach the new value.

L214-216: this explanation seems a little tentative. The abrupt features are super-imposed on the long-term precession-driven d18Oatm signal. Why would the earlier trend not just represent the orbital signal, for example? If abrupt transitions in the hydroclimate were involved, one would expect to see it in the Chines speleothems. Severinghaus et al. 2009 use the \Delta \epsilon_land to interpret d18Oatm, which is a more meaningful analysis.

L218: It seems a bit of a stretch to interpret the low-res Pa/Th data with such age uncertainty in this way. I think the authors need to exercise a bit more caution.

L226: Following Landais et al. (2013), we propose that. . ..

L242: where does the 70 ppb number come from? Based on the estimated degree of smoothing, what do you think the true magnitude of the DO event was like?

L248-249: I agree with this conclusion. Note that during the H1 CH4 event Chinese speleothems get more positive (and more negative for the 134 ka event).

L255: The magnitude of CH4 rises during DO events is modulated by the NH insolation signal. Based on that, can you place the 134 DO event into context for us? Are we expecting a large or big CH4 DO signal at 134 ka based on insolation?

L261: strengthening of the ASIAN monsoon system. . .. (probably SH tropics show opposite)

L285-292: the quenching of the emergent DO event by a MWP pulse does not explain the difference to the Bolling, because the Bolling was coincident with MWP 1A, which

seemed not to hurt it!

L302: As noted above, I think the short duration of the event is fully in line with expectations from DO events of the last glacial cycle, most notably DO2.

Fig 2 caption: what is the statement about AMOC vigor based on? And why are the transitions not aligned with those in the CH4 data? I think it would make more sense to align the boxes using your data. Also, shouldn't the newly discovered DO event have a red background (interstadial)?

Fig 4, right panel: why not show the d18O_atm here also? Would be very helpful. Can you show insolation somewhere, as well as Antarctic temperature? The slowdown in warming at 130.5 ka can be pointed out that way.

References:

Baumgartner, M., Kindler, P., Eicher, O., Floch, G., Schilt, A., Schwander, J., et al. (2014). NGRIP CH4 concentration from 120 to 10 kyr before present and its relation to a $\delta$15N temperature reconstruction from the same ice core. Clim. Past, 10(2), 903-920. http://www.clim-past.net/10/903/2014/

Buizert, C., & Schmittner, A. (2015). Southern Ocean control of glacial AMOC stability and Dansgaard-Oeschger interstadial duration. Paleoceanography, 30(12), 2015PA002795. http://dx.doi.org/10.1002/2015PA002795

Cheng, H., Edwards, R. L., Broecker, W. S., Denton, G. H., Kong, X., Wang, Y., et al. (2009). Ice Age Terminations. Science, 326(5950), 248-252. http://www.sciencemag.org/cgi/content/abstract/326/5950/248

Clark, P. U., He, F., Golledge, N. R., Mitrovica, J. X., Dutton, A., Hoffman, J. S., & Dendy, S. (2020). Oceanic forcing of penultimate deglacial and last interglacial sea-level rise. Nature, 577(7792), 660-664. https://doi.org/10.1038/s41586-020-1931-7

Epifanio, J. A., Brook, E. J., Buizert, C., Edwards, J. S., Sowers, T. A., Kahle, E. C., et

al. (2020). The SP19 chronology for the South Pole Ice Core - Part 2: gas chronology, delta age, and smoothing of atmospheric records. Clim. Past, 2020(16), 2431–2444. https://www.clim-past-discuss.net/cp-2020-71/

Oyabu, I., Kawamura, K., Kitamura, K., Dallmayr, R., Kitamura, A., Sawada, C., et al. (2020). New technique for high-precision, simultaneous measurements of CH4, N2O and CO2 concentrations; isotopic and elemental ratios of N2, O2 and Ar; and total air content in ice cores by wet extraction. Atmos. Meas. Tech., 13(12), 6703-6731. https://amt.copernicus.org/articles/13/6703/2020/

---

## Editor Comment (EC1) · Luke Skinner (Editor) · 9 Feb 2021

Dear Loic, if I may,

The review/discussion phase for your manuscript has now ended, and you are invited to respond to the review comments. I would like to add an editorial perspective, in case this helps to guide your response, and make the process more efficient.

Based on the review comments that have been received, I see there to be two major issues that must be addressed in a substantially revised version of your manuscript, before it can be considered for publication.

The first, emphasised primarily but not exclusively by Reviewer 2, is that your manuscript serves as presentation of an updated methodology, and yet appears to

fall short of what one might expect for a 'method paper'. As suggested by Reviewer 2, there are ways to do this without interrupting the flow of a paper that also seeks to present novel ideas and interpretations on palaeoclimate events and processes; however, I would suggest that one of these is to include a dedicated section that is clearly signposted after the introduction (e.g. "In section 2, we describe a revised method. . . Readers who are less concerned with the details of the methodology are invited to skip to section 3, where we discuss. . . etc. . . ."). I think the preference for a dedicated section versus an Appendix should be premised on the length of the text required, and whether or not the manuscript seeks to serve as a viable reference for the updated methodology. To me, this seems inevitable, because the new method has not been described elsewhere. A side note here, is the comment of Reviewer 1, that having developed and presented an updated methodology, a lack of confidence in the accuracy of the resulting data is suggested by the arbitrary correction that is subsequently applied – this deserves some discussion and context in a revised manuscript version.

The second major issue, emphasised primarily by Reviewer 1 this time, is that the main findings presented (the existence of centennial/millennial events within the HS11 'complex', and thus across TII, are not in themselves entirely new, and that some aspects of the interpretations appear to be somewhat confused (e.g. the concept of Heinrich-stadials, HS, merging into 'non-Heinrich, DO-, stadials' – opening up a terminological can of worms that is not resolved at all). My own feeling is that some confusion does indeed arise from the manipulation of concepts and terminology (generally from previous studies), while becoming somewhat detached from relevant observations (e.g. the end of a 'Heinrich event' would need to be defined in terms of ice-rafted debris deposition, etc. . .). In this respect, I find that Reviewer 1's comments have some traction, and should not be cast off lightly. My own suggestion is that the critique or Reviewer 1 could be addressed through a greater attention to relevant studies/datasets (for comparison), and perhaps also some effort placing key records on consistent age-scales etc. . . so as to show clearly the associations between e.g. AMOC, ocean oxygenation, North Atlantic ice-rafted debris deposition, terrestrial climate, sub-tropical hydrological

cycle, and polar ice-core archives, so as to make an observationally supported proposal for the mechanisms behind the events that you identify in your CH4 and N2O records. Here, I would like to emphasise that the key point made in your manuscript, regarding the existence of at least two different 'types' of sub-millennial event across TII and within the 'HS11 complex' (e.g. Tzedakis et al., NComms 2018), is sufficiently interesting and important to deserve being placed on a more robust observational footing.

In summary, it seems to me that the manuscript is to some extent caught in a position of tension, between serving as a 'method paper', and serving as a new investigation into sub-millennial variability across HS11 and TII. Furthermore, some work appears to be needed for the manuscript to meet either (or both) of these candidate goals. I therefore invite you to please consider preparing a significantly revised manuscript version that you believe addresses the issues raised by both reviewers, and that will likely be reviewed by a third and independent expert with access to the first set of comments. I further suggest that you explicitly take into account the detailed comments of the two reviewers in any revised manuscript. I sincerely hope that you will see this, as I do, as an opportunity to revise your study in such a way as to make it significantly more impactful. I believe that with some further analysis, and redrafting of figures, this manuscript could indeed serve as a key reference that highlights the importance and possible significance of late glacial/deglacial variability across TII.

Sincerely, Luke Skinner

---

## Author Comment (AC1) · 5 Mar 2021

**Final author response**

March 5, 2021

Dear reviewers, dear editor

We would like to thank the two anonymous reviewers and the editor for having carefully commented on our manuscript. We respond to the remarks below, using the following color code:

- Original text from the reviewers and editor in black

- Our responses in blue

- Proposed modifications in green

**1 Response to the editor**

The review/discussion phase for your manuscript has now ended, and you are invited to respond to the review comments. I would like to add an editorial perspective, in case this helps to guide your response, and make the process more efficient.

Based on the review comments that have been received, I see there to be two major issues that must be addressed in a substantially revised version of your manuscript, before it can be considered for publication. The first, emphasised primarily but not exclusively by Reviewer 2, is that your manuscript serves as presentation of an updated methodology, and yet appears to fall short of what one might expect for a 'method paper'. As suggested by Reviewer 2, there are ways to do this without interrupting the flow of a paper that also seeks to present novel ideas and interpretations on palaeoclimate events and processes; however, I would suggest that one of these is to include a dedicated section that is clearly signposted after the introduction (e.g. "In section 2, we describe a revised method ... Readers who are less concerned with the details of the methodology are invited to skip to section 3, where we discuss ... etc. ..."). I think the preference for a dedicated section versus an Appendix should be premised on the length of the text required, and whether or not the manuscript seeks to serve as a viable reference for the updated methodology. To me, this seems inevitable, because the new method has not been described elsewhere. A side note here, is the comment of Reviewer 1, that having developed and presented an updated methodology, a lack of confidence in the accuracy of the resulting data is suggested by the arbitrary correction that is subsequently applied – this deserves some discussion and context in a revised manuscript version.

**Response 1**:

The instrument used in our study is an improved version of the previous device used in Bern. It combines established techniques (gas chromatography and continuous extraction under vacuum), commonly used in ice core measurement systems (e.g. Baumgartner et al., 2014; Flückiger et al., 2004; Schilt et al., 2010a,b; Schmitt et al., 2014). Therefore, we feel that the overall degree of novelty does not legitimate a stand-alone method paper. Motivated by the comment of reviewer 2 (R2), we will provide a more thorough description of the analytical system (please see response 2 to R2). We are convinced that a dedicated method chapter, either in the main text or the appendix (depending on the length), is an appropriate format to provide a concise, yet comprehensive description of the system. In case the method section is kept in the main text we will include a guideline to skip this part, as suggested by the editor, intended for the uninterested reader.

We understand that the confidence in the accuracy of the results constitutes a major issue. We provide hereafter a comprehensive discussion on this (please see response 1 to R1).

The second major issue, emphasised primarily by Reviewer 1 this time, is that the main findings presented (the existence of centennial/millennial events within the HS11 'complex', and thus across TII, are not in themselves entirely new, and that some aspects of the interpretations appear to be somewhat confused (e.g. the concept of Heinrich- stadials, HS, merging into 'non-Heinrich, DO-, stadials' – opening up a terminological can of worms that is not resolved at all). My own feeling is that some confusion does indeed arise from the manipulation of concepts and terminology (generally from previous studies), while becoming somewhat detached from relevant observations (e.g. the end of a 'Heinrich event' would need to be defined in terms of ice-rafted debris deposition, etc. . . ). In this respect, I find that Reviewer 1's comments have some traction, and should not be cast off lightly. My own suggestion is that the critique or Reviewer 1 could be addressed through a greater attention to relevant studies/datasets (for comparison), and perhaps also some effort placing key records on consistent age-scales etc. . . so as to show clearly the associations between e.g. AMOC, ocean oxygenation, North Atlantic ice-rafted debris deposition, terrestrial climate, sub-tropical hydrological cycle, and polar ice-core archives, so as to make an observationally supported proposal for the mechanisms behind the events that you identify in your $CH_4$ and $N_2O$ records. Here, I would like to emphasise that the key point made in your manuscript, regarding the existence of at least two different 'types' of sub-millennial event across TII and within the 'HS11 complex' (e.g. Tzedakis et al., 2018), is sufficiently interesting and important to deserve being placed on a more robust observational footing.

**Response 2**:

It is true that the existence of centennial/millennial $CH_4$ and $N_2O$ fluctuations across TII was already speculated before but the resolution of existing records did not allow for an unambiguous answer. We elaborate on this hereafter (please see response 2 to R1). Moreover, we acknowledge that we have indeed confused the issue by introducing misleading terminology (e.g. the HS stadial becoming a DO-stadial. . . ). We address this carefully in our response (please see response 3 to R1, response 4 to R1 and response 6 to R2).

We agree with the editor that consistent age-scales constitute a prerequisite for a meaningful interpretation. The records we include in our analysis are either on (or synchronized with) the Antarctic Ice Core Chronology 2012 (AICC2012) (Bazin et al., 2013; Veres et al., 2013) or on the absolute radiometric age-scale. For each comparison involving a feature on the radiometric timescale and a feature on the AICC2012, we take into account the respective dating uncertainty to asses whether two events occurred simultaneously. A possible step further would be to transfer the AICC records on the absolute radiometric age-scale. For such a synchronization, we need tie points independent from the results of our study. Since the 129-ka event (formerly referred to as the 128-ka event, please see response 5 to R1) was already known before (i.e, it does not constitute a feature newly resolved by our study), it would be possible to match $CH_4$ and $\delta^{18}O_{calcite}$ there, assuming that monsoon changes and northern tropical $CH_4$ emissions are tightly linked. However, this would constitutes only a single tie point, which is likely insufficient to perform a robust synchronization.

In summary, it seems to me that the manuscript is to some extent caught in a position of tension, between serving as a 'method paper', and serving as a new investigation into sub-millennial variability across HS11 and TII. Furthermore, some work appears to be needed for the manuscript to meet either (or both) of these candidate goals. I therefore invite you to please consider preparing a significantly revised manuscript version that you believe addresses the issues raised by both reviewers, and that will likely be reviewed by a third and independent expert with access to the first set of comments. I further suggest that you explicitly take into account the detailed comments of the two reviewers in any revised manuscript. I sincerely hope that you will see this, as I do, as an opportunity to revise your study in such a way as to make it significantly more impactful. I believe that with some further analysis, and redrafting of figures, this manuscript could indeed serve as a key reference that highlights the importance and possible significance of late glacial/deglacial variability across TII.

**2 Response to reviewer 1 (R1)**

This manuscript describes a new protocol for the measurement of $CH_4$ and $N_2O$ concentrations on small samples of ice. The new method is applied to samples from the EDC ice core and enables new composite records to be developed covering the penultimate deglacial period (Termination 2).

I'm sorry to say that I was rather underwhelmed by this paper. I had expected to see exciting new records with new insights and an opportunity to learn more about T2. Indeed, there are a couple of novel features reported but nothing very exciting. In all I suspect this paper will add to a growing body of studies dealing with T2 that will ultimately (but not yet) lead to an increase in understanding.

The authors describe a new analytical approach, an important step forward which deserves to be documented. However, the authors point to a significant offset with measurements made by earlier methods and end up correcting their new data in an ad hoc fashion (minimising the difference between the various datasets). This operation implies that the authors have little confidence in the absolute values of their results, and this is obviously alarming. Is there really no way to produce a standard that can be used to cross calibrate between techniques?

**Response 1**:

It is not possible to produce artificial "standard" ice core samples containing air with calibrated trace gas concentrations relevant for paleoclimate reconstructions. For this reason, it is generally hard, if not impossible, to prove which extraction technique provides the most accurate results. One approach would be to measure samples from the same depth levels within the same ice core (assuming identical trace gas concentrations) with different setups and compare the results. In our case, this approach is not possible as the former instrument used in Bern to acquire the EDC data was destroyed by a fire in the laboratory (this was also the reason to build the new system presented here). It is therefore not possible to cross calibrate between the old and new setups in our laboratory. In the end, such a cross-calibration would also lead to an "ad-hoc" correction as we do not know which instrument delivers the most accurate results. At this point, it should be stressed that our new data show the same relative variations as the existing datasets but a significant offset on the order of a few percent of the absolute concentrations. Note that previous $CH_4$ data from different laboratories (using similar melt-refreeze extraction techniques) also differed by $\sim$5-10 ppb and used an offset correction to bring the records on the same scale. The fact that we correct our new data does not reflect our insufficient trust but constitutes an effort to stay on consistent $CH_4$ and $N_2O$ scales.

Motivated by the comments of the reviewers, we reassessed our measurement and evaluation schemes to examine whether a part of the offset compared to the old data can be explained by our procedure. We found that part of the observed offset has been caused by our evaluation scheme. We updated this evaluation scheme which now avoids an amount dependency correction. The evaluation is now simpler and more reliable. Moreover, we found a mathematical error in the $CH_4$ line offset correction function we were previously applying. Overall, our $CH_4$ data are now lower, resulting in an offset compared to the old data of $18 \pm 10$ ppb (instead of $29 \pm 7$ ppb, a reduction of nearly 40%). The $N_2O$ data has not changed substantially and the offset appears now, if at all, slightly increased ($21 \pm 3$ ppb instead of $18 \pm 2$ ppb). The re-evaluated data will be included in our revised manuscript.

It seems likely that the majority of the offset is linked to the different extraction techniques. The published EDC data were measured with a melt-refreeze technique, where the amount of gases trapped in the refrozen meltwater was quantified by expanding standard gases over gas-free ice samples during a melt-refreeze cycle. This approach may not be adequate because the relative amount of air and trace gases dissolved in the refrozen meltwater may not be the same for gas-free samples and ice core samples (Ryu et al., 2018). Procedures involving the repetition of the melt-refreeze cycle and analyzing the residual gas are currently preferred (e.g. Lee et al., 2020; Ryu et al., 2018; Yang et al., 2017). Although we acknowledge that a 18 ppb offset appears to be relatively large, it is not surprising that our extraction technique leads to overall higher values as the extracted gas is quantitatively removed from the extraction chamber.

As a conclusion, we do not know which instrument delivers the most accurate values. Because melt-refreeze has been the defacto standard procedure for decades and because the EDC records have been measured with these systems, we think additional overlapping measurements with an independent extraction technique (i.e sublimation) are needed to resolve the dispute. In the meantime, and without further evidence, we prefer staying on the safe side and scaling our data to the

existing EDC benchmark record. We will elaborate on this in the revised manuscript to justify our approach.

The data themselves are interesting but offer little new insight beyond increasing temporal resolution of older records from the same ice core. The new composite record of $N_2O$ does improve on the older record but the $CH_4$ record merely confirms that details which previously were suspected are actually real.

**Response 2:**

It is true that the existence of centennial/millennial $CH_4$ and $N_2O$ fluctuations across TII, especially at ∼134 ka BP, was already speculated before. However, the resolution of the previous EDC data (nor the resolution of the other deep Antarctic ice cores) was far from being sufficient to unequivocally prove the existence of the features at 134 and 130.5 ka BP (Buiron et al., 2011; Chappellaz et al., 1990; Loulergue et al., 2008; Schilt et al., 2010a,b; Sowers, 2001). As an example, the 134-ka event was recorded in a single $CH_4$ and $N_2O$ data point in the EDC ice core (Loulergue et al., 2008; Schilt et al., 2010a) and two $N_2O$ data points in the Vostok ice core (Sowers, 2001). The unveiling of the fluctuations at ∼134 and ∼129 ka BP can therefore be considered as a novelty of this work.

Some confusing nomenclature is developed here that leads to ambiguity and a loss of logic. For example, the authors distinguish 'late stadial' from 'intra-stadial' variability, which is fine on the face of it but becomes confused when they discuss the 134ka event, which occurs within a stadial event (HS11) but is apparently not an 'intra-stadial' event (somewhat of an oxymoron?). This confusion comes from the fact that the authors are using a non-specific term (intra-stadial) to define a specific mode of variability that was described in a paper(s) by Rhodes et al. (2015) and was previously argued to be related to strengthening of southern hemisphere monsoon systems and a southward shift of the ITCZ (as opposed to a northward shift, which might be expected with the abrupt transitions from stadial to interstadial state). Perhaps the authors need to find an alternative (more descriptive) name for these 'type' of event.

**Response 3:**

We agree with the referee that the choice of wording was confusing. We propose the following nomenclature throughout the revised manuscript:

- "DO-type" for $CH_4$ and $N_2O$ fluctuations concomitant with the transition from a stadial to an interstadial, also suggested by reviewer 2.

- "HS-type" (instead of "intra-stadial") for the $CH_4$ fluctuations occurring within Heinrich Stadials (Rhodes et al., 2015), also suggested by reviewer 2.

- "late HS-type" (instead of "late stadial") for the $N_2O$ increase at the end of the Heinrich Stadials (Fischer et al., 2019; Schilt et al., 2013, 2014).

More confusion occurs with the discussion of the 130.5ka 'event', which the authors suggest might represent a transition from an HS event to a D-O stadial. This description doesn't make much sense to me I'm afraid. A Heinrich-stadial (HS) has been defined as a stadial that contains a Heinrich event (HE). Thus the label HS11 implies that this cold interval contains the Heinrich event HE11. It makes no sense to imply that HS11 can change to a regular stadial (that does not contain an HE) once Heinrich Event 11 has ended (if that is what happens). I suppose you could argue that HE11 (Heinrich Event 11) ended before the end of HS11 but HS11 does not become a regular stadial once the HE is over. I realise that the present authors took this idea from an earlier paper by Landais et al. (2013) but it makes no more sense in that paper.

**Response 4:**

We thank the reviewer for this important criticism. We will abandon the section of the text mentioning that a DO stadial occurred within HS11. In fact, we also decided to remove the entire section of the discussion where the DO-stadial is mentioned (L210-228). Indeed, relating the partial AMOC resumption and the small intensification of the hydroclimate to the late-HS type $N_2O$ increase appears too speculative. Instead, we will only refer to Schmittner and Galbraith (2008), who related this mode of $N_2O$ variability to long-term adjustments of the nitrate and oxygen inventories in the upper-ocean. Finally, the label "stadial" will also be removed from Fig. 2 and 3.

Perhaps I am missing something obvious but the authors' discussion and figures describe an event at 128ka which I believe occurred between 128.6 to 128.7ka, in which case they should perhaps round up to 129ka rather than down to 128ka. Rounding to the nearest kyr and adding a tilde would be fine in the text ($\sim$129ka) but this is not something to do in a figure.

**Response 5:**

We agree and will round to 129 ka BP and will refer to this event as the "129-ka event" throughout the revised manuscript.

In figures 2 and 3, the authors indicate a transition from stadial to interglacial conditions at 128ka that does not align with the rapid increase in $CH_4$ that would more commonly be interpreted to indicate the end of a stadial period (cf end YD as shown in their Fig. 2).

**Response 6:**

We will make sure that the end of every stadials in Fig. 2 and 3 (transition HS1 to BA, YD to Holocene and HS11 to LIG) is aligned with the rise in $CH_4$ (this is also a remark of reviewer 2). We propose to replace the colored boxes by vertical lines delineating the climate periods. Motivated by reviewer 2 we also propose to show the 134-ka event in Fig.2 and 3 (label + vertical bar).

My final major gripe concerns the authors' discussion and interpretation of their results. Most of the interpretation follows previous studies (e.g. rapid CH4 rise coincides with AMOC recovery at end of a stadial period) but where there is something new to discuss (e.g. the mysterious 134ka event) the authors provide a very slim argument for a major conclusion within their abstract (that this event was analogous to a D-O warming of MIS 3).

**Response 7**:

We politely disagree with this comment. Our argumentation, that the 134-ka event in $CH_4$ and $N_2O$ is indeed a "DO-type" fluctuation (an opinion also shared by reviewer 2), does not appear slim to us. In fact, we provided several independent lines of evidence (simultaneous $CH_4$ and $N_2O$ rises, magnitude and duration of the fluctuations and concomitant northward shift of the ITCZ). A clear AMOC change is the only missing piece of evidence to confirm our argumentation. Moreover, the low resolution marine records does not allow us to exclude the existence of such an AMOC resumption either. Finally, we do not think that our abstract particularly highlights the 134-ka event.

Below I outline some more comments in order of their appearance in the ms.

Abstract Line 5/6: "These features occurred in concert with reinvigorations of the Atlantic Meridional Overturning Circulation (AMOC) and northward shifts of the Intertropical Convergence Zone." Implies that the authors have identified at least 2 instances of this feature, which turns out to be a single instance (which was previously known about) and an additional instance that is substantially unsupported.

**Response 8**:

We agree that there is only one unequivocal instance of AMOC reinvigoration (at the transition to the LIG).

L5-6: We will remove the sentence "These features occurred in concert with reinvigorations of the Atlantic Meridional Overturning Circulation (AMOC) and northward shifts of the Intertropical Convergence Zone."

Line 8: (editorial) The authors' use of ka and ka BP needs attention; ka is fine on its own as shorthand for kyr ago or kyr BP (avoid ka ago or ka BP).

**Response 9**:

We politely disagree with this comment. It is common in the field to use ka BP as a shorthand for thousand years before present (e.g. the age-scale paper of Buizert et al. (2015)). We introduce this definition both in the abstract and in the main text.

Line 8 and throughout: ". . .are assimilated to. . .". The meaning here is not clear – please check use of English throughout.

**Response 10:**

We will replace "assimilated to" by "resemble".

L7-10: The revised sentence now reads: "The abrupt $CH_4$ and $N_2O$ rises at 134 and 128 thousand of years before present (hereafter ka BP) resemble the fluctuations accompanying the Dansgaard-Oeschger events of the last glacial period, while rising $N_2O$ levels at 130.5 ka BP resembles the pattern of increasing concentrations that characterized the end of Heinrich stadials".

Main text Line 45: 'late stadial' – perhaps better as 'late HS'?

**Response 11:**

We agree (please see response 3).

Line 46: "This mode of variability has been evidenced for the HS during the last glacial period. . ." Please check English.

**Response 12:**

We agree.

L46-47: The revised sentence now reads: "This mode of variability characterized the HS during the last glacial period (Schilt et al., 2013) and the last deglaciation (Fischer et al., 2019; Schilt et al., 2014)".

Line 66-75: Much of this intro section feels more like discussion.

**Response 13:**

We present here the environmental controls modulating $N_2O$ sources based on the literature review of pre-existing results. We wrote a similar section for $CH_4$ at L55-63. It seems consistent to introduce these environmental controls also for $N_2O$, as some of this knowledge will later be used in the discussion.

Results section: Much of the text here is unnecessary and feels verbose; you don't need to describe every inflection and quantify every rate - you have figures for this!

**Response 14**:

We propose to explicitly mention in the text the values essential to the reader. All other values will be shown in a supplementary table in the appendix. These numbers (rates of change, amplitude, duration, etc.) will be reevaluated with the revised dataset (please see response 1).

L119: The revised section now reads: "Our records display several fluctuations standing out in the overall evolution of the $CH_4$ and $N_2O$ concentrations (Fig. 1). Abrupt $CH_4$ rises are identified at ∼134 and ∼129 ka BP. At 134 ka BP, concentrations increased by ∼70 ppb in ∼200 years before declining by ∼50 ppb in ∼200 years. The 129-ka event consists of a ∼190 ppb increase (about half of the deglacial change) proceeding in ∼300 years.
The evolution of $N_2O$ concentrations alternates between periods of plateaus and well-marked fluctuations (Fig. 1). Similar to $CH_4$, a feature is resolved at ∼134 ka BP where concentrations increased by ∼30 ppb in ∼200 years before stabilizing during ∼200 years and declining in ∼1000 years. The 129-ka event is also imprinted in our $N_2O$ record and is characterized by a ∼30 ppb rise in ∼800 years. In addition, an increase is identified at ∼130.5 ka BP, where concentrations rose by ∼20 ppb in ∼700 years. The 130.5-ka and 129-ka events are separated by a plateau that lasted ∼800 years".

Line 129: "Abrupt $CH_4$ rises are identified at ∼134 and ∼128 ka BP" Again, if rounding is to be used then rounding up from 128.6/7 to 129 ka would be more conventional.

**Response 15**:

We agree (please see response 5).

Discussion Line 154: But the $CH_4$ point at 139.9ka is from a previous study yes? Need to make this clearer.

**Response 16**:

We agree.

The revised sentence now reads: "The $CH_4$ data point at 139.9 ka BP, previously published by Loulergue et al. (2008), is measured in the section characterized by the widest GAD in our record".

Line 210: ". . .remarkably coeval. . .". Nothing remarkable here, perhaps approximately coeval.

**Response 17**:

We agree and will remove "remarkably".

Line 229: "Overall, the 130.5-ka event fits into the framework of the late stadial events (Fischer et al., 2019; Schilt et al., 2013, 2014) and can be viewed as an analogue of the late HS1 rise during TI." I am not convinced by this; the rise in $N_2O$ ~130.5 ka is very similar in magnitude and rate as that ~129ka and yet the 'late stadial' rises documented for T1 and MIS 3 (Schilt et al., 2013) are significantly slower than the rather abrupt rise that accompanies the transition to interstadial conditions. Given that the 130.5ka rise is also separated from the 'end-stadial' rise by ~1000 years, which is also not noted for events during T1 or MIS 3 I think there is room for questioning the validity of this analogue.

**Response 18**:

We agree that the $N_2O$ rise at 130.5 ka BP differs from the typical "late HS-type" of fluctuation based on the rate of change and on the temporal decoupling with the following "DO-type" of increase. However, it also shares strong similarities (timescale, absence of $CH_4$ concomitant change, occurrence at the later part of a HS). We will soften the conclusion that the 130.5-ka event fits into the framework of the late-HS type of fluctuations. Please note that the rates of change will be reevaluated with our revised dataset (please see response 1).

The revised sentence now reads: "Despite the different rates of change, the 130.5-ka event likely constitutes an instance of a late-HS event during TII".

Lines 235-284: I agree that something interesting happened ~134ka and that the $CH_4$ and $N_2O$ evidence presented here, together with the speleo records and monsoon reconstruction reported by Nilsson-Kerr et al. (2019), suggest that something atmospheric was involved. The logical next step is to invoke a change in AMOC but there is no firm evidence for this. The authors argue (reasonably) that the age models employed for ODP 1063 by Böhm et al. (2015) and Deaney et al. (2017) are uncertain enough to allow significant wiggle room but the change in $\varepsilon_{Nd}$ that would involve (up to 2 epsilon units) is nearly half of that associated with the hypothesised AMOC resumption ~129ka. If eNd is taken at face value as a deep-water circulation tracer (which is not entirely free of problems) then why do we not find more evidence of such a large event ~134ka throughout the North Atlantic? Furthermore, the authors state that the timescale of $CH_4/N_2O$ change across this event 'precludes' an oceanic source – so why call on an oceanic mechanism? Perhaps the authors need to soften their argument that the 134ka event is really such a good analogue for a D-O event (see earlier comment on Lines 5/6 of abstract).

**Response 19**:

We mention an oceanic mechanism because "DO-type" of events are typically associated with AMOC changes, not to say that $N_2O$ is released from the ocean. Beside the absence of a clear and unequivocal AMOC signal, we think the 134-ka event can reasonably be considered as an instance

of a "DO-type" of event (an opinion also shared by reviewer 2). We are, as of now, not aware of other AMOC proxies registering an imprint of the 134-ka event. Given its short duration, marine records with millennial resolution may not capture it.

The figures and fonts on these are painfully small.

**Response 20**:

We will increase their sizes.

**3    Response to reviewer 2 (R2)**

Schmidely et al. provide new high-resolution $CH_4$ and $N_2O$ datasets over the penultimate deglaciation from the EDC ice core, and based on their data suggest an interpretation of the last deglaciation that has more structure and richness that the "canonical" interpretation of just a long AMOC shutdown over H11 from 136 to 128 ka BP (Cheng et al., 2009; Clark et al., 2020). While the interpretation is somewhat speculative at times, I for one believe that they are mostly correct. It is always exciting to see that the climate is more complicated than we thought at first - it always pays off to evaluate ice core records at the highest possible resolution. I think the manuscript is suitable for publication after addressing some issues. In particular, the new analytical setup (a major achievement by itself) deserves some more attention, as well as the observed (and surprisingly large) offset with existing records.

(1) The 134 ka DO event

I am most excited about the discovery of a minor DO event early in the deglacial sequence (134 ka BP). When first looking at their data, I assumed the 134 ka $CH_4$ features was the TII equivalent of the 16.2 ka Heinrich 1 $CH_4$ feature identified by Rhodes et al. (2015). However, based on the Chinese speleothem record, as well as the size of the feature, Schmidely et al. convincingly argue that it is in fact a small DO event. Personally, I think the authors should present this 134ka feature as an analog of DO2 – which likewise has an extremely short duration, and a (very!) small $CH_4$ peak. The two have a similar orbital configuration, with very small NH summer insolation.

The small magnitude of the CH4 peak seems in line with the small NH insolation at 134 ka (Baumgartner et al., 2014).

The authors struggle to explain the short duration of the event relative to the Bolling, but I think this is not necessarily hard. It has been shown (Buizert and Schmittner, 2015) that the duration of DO events scales strongly with Antarctic temperature (and following more recent work, presumably also with mean ocean temperature). The 134 ka DO event occurred much earlier in the TII deglacial sequence than the Bølling event in the TI sequence. Therefore, during the 134ka DO event global $CO_2$ and Antarctic temperature were both lower than they were during the Bølling. These factors

would predict a much shorter DO event at 134 ka than at 14.7 ka, as is indeed the observation of the authors.

The idea that meltwater "squelched" the DO event seems unlikely to be the complete explanation by itself, given that the Bølling was likewise accompanied by MWP-1A. Such a MWP is indeed perhaps the result of suddenly warming the NH via AMOC invigoration, thereby melting ice at the NH high-latitudes. However, Buizert and Schmittner (2015) show that under colder Southern Ocean background conditions (as during the 134 ka event), the AMOC is much more susceptible to meltwater perturbation. This may explain why the Bolling survived MWP-1A, and yet the 134ka event did not survive MWP-2B.

**Response 1:**

We will deepen the analysis of the amplitude and duration of the 134-ka event to anchor it better in the frame of the "DO-type" of events.

- We will analyze the amplitude of the $CH_4$ fluctuation at 134-ka in the light of insolation (Brook et al., 1996; Flückiger et al., 2004). We propose to add a plot showing average low-latitude summer insolation (Berger, 1978) versus amplitude of "DO-type" $CH_4$ features for the past 145 thousand years BP (using the WDC data of Rhodes et al. (2015) for the last 67 thousand years BP and our data for the time period 125-145 ka BP). Since we aim at comparing features recorded in the EDC and WDC ice cores, we will retrieve the unsmoothed atmospheric $CH_4$ signal from our data using the approach of Nehrbass-Ahles et al. (2020).

- We will analyze the duration of the $CH_4$ fluctuation at 134 ka BP in the light of temperature in the high-latitude Southern Hemisphere (Buizert and Schmittner, 2015). We propose to add a plot showing Antarctic temperature (using the Antarctic temperature stack of Parrenin et al. (2013), converted to the AICC timescale) versus duration of "DO-type" $CH_4$ features (using again the WDC data of Rhodes et al. (2015) for the last 67 thousand years BP and our data for the time period 125-145 ka BP).

We agree that the 134-ka event is a close analogue of DO2. However, we decided to refrain from drawing strict analogies between individual events (e.g. the 128-ka event is an analogue of the end of the YD or the BA, the 134-ka is an analogue of DO2...). Such analogies are disputable. We will focus our efforts on relating the features we resolved to recurrent modes of variability ("DO-type", "HS-type"...) and showing that these modes are also observable during the last deglaciation. We believe that avoiding strict analogies between single events will enhance the overall clarity of the text.

We propose the following modification in the text:

L285-303 : We will remove "Should our interpretation hold, [ . . . ] we are currently unable to propose a mechanism accounting for the relative brevity of the 134-ka event."

The revised text now reads: "Should our interpretation hold, the 134-ka event can be considered as a short DO-event, where the hypothesized AMOC reinvigoration might have been perturbed by

Meltwater Pulse 2B (MWP-2B). MWP-2B represents $\sim$70 % of the deglacial sea level change and coincides with the 134-ka event within dating uncertainty (Marino et al., 2015). The disruption of the AMOC by freshwater forcing might have been enabled by the high susceptibility of the AMOC to perturbations at time the southern high-latitudes are particularly cold (Buizert and Schmittner, 2015), as was the case during HS11."

(2) The analytical setup

The new setup is a major part of the paper, and has not been documented elsewhere. The line uses some new methodologies that have not previously been used for $CH_4/N_2O$ analysis. It thus seems necessary to provide more details on this aspect of the paper. The authors should provide a schematic drawing of the line, and discuss its operation in more detail. For example, how is the sample calibration done, and how often? What is the precision of the new line from replicates? Do you calculate concentration using the air concentration from the thermal conductivity detector? What carrier gas is used? What is the "line offset"? How many samples can you analyze? Do you apply a solubility correction? Does the line produce total air content data?

Most importantly, the authors should perform a more thorough investigation of the large offset of the new setup with old data. The authors choose to correct the new data, implying they trust the old method better. What could cause such an offset? It would need to be understood if the setup is to be applied to periods of time where we do not have a reliable $CH_4$ record already that can be used for offset correction – during such periods we need to be able to rely on the internal calibration of the line.

Perhaps all this material could be place in an appendix as to not affect the flow of the paper. This would be a valuable place of reference for future work that uses this setup.

**Response 2:**

We agree with the reviewer's comment and will provide a comprehensive description of the analytical system. This will include, among others, a figure of the flow scheme, description of the measurement and evaluation procedures as well as key information regarding the performance of the instrument (sample throughput, line offset, precision...). Depending on the length of the revised text we will keep it as a section in the main text, with an invitation to skip for the uninterested reader (as suggested by the editor), or include it in the appendix. The offset relative to the old data is indeed a major point (please see response 1 to R1).

(3) Line-by-line comments:

L8 and L9: What does "assimilate" mean in this context? Please clarify

**Response 3**:

We will change the wording (please see response 10 to R1).

L13: why compare it to the Bølling? Why not DO2 – based on background climate (insolation, $CO_2$, mean ocean temp, etc) that may be a closer analog. Compared to DO2 it looks quite normal.

**Response 4**:

We agree (please see response 2).

L34: What is the width of the gas age distribution using the recent estimate of Epifanio et al. (2020) who related it to Delta-age?

**Response 5**:

According to Epifanio et al. (2020), the width of the gas age distribution amounts to 3% of the delta age. Using the minimum and maximum delta age values for our study period gives a width ranging between ∼50 and ∼130 years. With the approach of Nehrbass-Ahles et al. (2020), using the minimum and maximum delta values give mean widths ranging between ∼100 and ∼220 years for our study periods.

L32: The revised text now reads "We increased the sampling density of the aforementioned records by a factor ∼3.5 and ∼5 to obtain mean resolutions of 100 and 115 years for $CH_4$ and $N_2O$, respectively. These values are on the order of the width of the gas age distribution (GAD) at EDC ice for the time interval 145-125 ka BP. The mean width of the GAD is in the range ∼100-220 years (using the approach of Nehrbass-Ahles et al. (2020), where the width is defined as the arithmetic mean of the distribution) and ∼50-130 years (using the approach of Epifanio et al. (2020), where the width is defined as the spectral width of the distribution)."

Page 2: The distinction between millennial and centennial $CH_4$ events is somewhat arbitrary and perhaps even incorrect. Several of the DO events last only a few hundred years, and some of the H-event $CH_4$ changes persist > 1000 years (like H4 and H5). Maybe just call them DO and H-event $CH_4$ features? That would be much clearer.

**Response 6**:

We agree that referring to the events based on their durations can be misleading. We propose a more consistent nomenclature for the events (please see response 3 to R1).

L61: "believed to" should be "hypothesized to"

**Response 7**:

L61: The revised sentence now reads: "On the other hand, HS-type of fluctuations are hypothesized to result from southward shifts of the ITCZ, strengthening monsoonal precipitation in the Southern Hemisphere (SH) tropics, leading to an increase in wetland emissions there (Rhodes et al., 2015)."

L73 is "though to be" driven exclusively by . . .

**Response 8**:

L73: The revised sentence now reads: "Finally, the late HS-type of $N_2O$ increases during TI is thought to be driven exclusively by marine emissions (Fischer et al., 2019; Schilt et al., 2013, 2014), maybe resulting from a long-term reorganization of the nitrate and oxygen concentrations following the preceding AMOC collapse (Schmittner and Galbraith, 2008)".

L78: maybe give a one-sentence description of the IRMS setup for $N_2O$ isotopes

**Response 9**:

We will include this in the introduction.

The revised text now reads: "The measurements of $CH_4$ and $N_2O$ concentrations were performed with a newly developed analytical system, firstly deployed for this campaign. The $\delta^{15}N(N_2O)$ and $\delta^{18}O(N_2O)$ data were measured with the device described in Schmitt et al. (2014), combining continuous extraction under vacuum with gas chromatography (GC) and isotope ratio mass spectrometry."

L91: acknowledge this is also the approach used by the Japanese lab at NIPR (Oyabu et al., 2020)

**Response 10**:

We will acknowledge the work of Oyabu et al. (2020) in the revised method section.

Section 2: More details are needed here, as outlined above.

**Response 11**:

We agree (please see response 2).

L102: This is a very large offset – surely much greater than the specified precision of either line. Can you clarify? The approach is quite cavalier. An explanation of the offset is needed. Is this due to the new method? That is implied by the correction approach. The comparison in Fig. 1 implies that the offset may be concentration-dependent (both $CH_4$ and $N_2O$). Can you show a scatter plot of old vs. new (found with the spline method), with a linear fit? It appears it may not be a constant offset as applied.

**Response 12:**

We agree (please see response 1 to R1). We will investigate a possible dependence of the offset on the concentrations and include the suggested scatter plot.

L147: hidden in the previously published?

**Response 13:**

Yes, we will make this clearer.

L147-148: The revised sentence now reads: "Overall, the improved resolution of our records allowed us to identify features not resolved in the previously published $CH_4$ and $N_2O$ EDC datasets".

L152: could you add subheadings to the discussion? This is just one long block of text now. Maybe separate out the discussion of the 130.5 ka event and the 134 ka event, for example. There may be other subheadings that can be added. This would help structure the discussion better.

**Response 14:**

We propose to split the discussion into four subheadings:

- Non-atmospheric $CH_4$ and $N_2O$ variability (L153-183)

- The 129-ka event (L184-204)

- The 130.5-ka event (L205-234)

- The 134-ka event (L235 onwards)

L155: what does "tentative approach" mean?

**Response 15:**

This is an inadequate wording for "empirical", we will delete "tentative".

L155: How do you define the width? Please be specific. Some common metrics are the second moment of the distribution (or variance, or spectral width) or the FWHM (full width at half maximum). Just be specific how you define the width.

**Response 16:**

The definition of the width will be mentioned earlier in the revised manuscript (please see response 5).

L155: The revised text now reads: "The $CH_4$ data point at 139.9 ka BP, previously published by (Loulergue et al., 2008), is measured in the section characterized by the widest GAD in our record, on the order of $\sim$220 years (using the approach of Nehrbass-Ahles et al. (2020)) or $\sim$120 years (using the approach of Epifanio et al. (2020)) for the time interval 141-139 ka BP. At 139.9 ka BP, the adjacent data points are 160 years older and 173 years younger than the dubious measurement. Consequently, it appears suspicious that such a large fluctuation would only be recorded in a single data point. In the following, we consider this measurement as an outlier, more likely resulting from the analytical procedure than representing atmospheric variability".

L185: the 128 ka may as well be an analogue of the Bølling transition that ended HS1.

**Response 17:**

We agree (please see response 1).

L189: It looks like the 128ka event in $CH_4$ is more similar to the Bølling transition in terms of magnitude and timing.

**Response 18:**

We agree (please see response 1).

L190: I doubt the difference in magnitude is related to smoothing, since the step in $CH_4$ lasts for millennia, so that both sites reach the new value.

**Response 19:**

We did not mean that the difference in magnitude is related to smoothing. The difference in magnitude is due to the lower concentrations at the end of the YD before the rise. We think the shorter timescale and the overshoot may be partly due to smoothing. Since it may be confusing and does not add to the discussion, we decided to remove this sentence (L189-192).

L214-216: this explanation seems a little tentative. The abrupt features are super- imposed on the long-term precession-driven $\delta^{18}O_{atm}$ signal. Why would the earlier trend not just represent the orbital signal, for example? If abrupt transitions in the hydroclimate were involved, one would expect to see it in the Chinese speleothems. Severinghaus et al. (2009) use the $\Delta\varepsilon_{land}$ to interpret $\delta^{18}O_{atm}$, which is a more meaningful analysis.

**Response 20:**

The interpretation of the small drop in $\delta^{18}O_{atm}$ as reflecting a change in the low-latitude hydroclimate is taken directly from Landais et al. (2013). The argumentation is strengthened by the concomitant small drop in $\delta^{18}O_{calcite}$ of SB25 speleothem (Cheng et al., 2009) at $\sim$130.5 ka BP followed by a plateau (light green curve in Fig. 3 (C)), resembling the structure observed in the $\delta^{18}O_{atm}$ record. We agree that $\Delta\varepsilon_{land}$ is a more robust proxy for past changes in the low-latitude hydrological cycle driven by ITCZ changes (Seltzer et al., 2017; Severinghaus et al., 2009).

We decided to remove the part of the discussion where $\delta^{18}O_{atm}$ is included (L210-228) (please see response 4 to R1).

L218: It seems a bit of a stretch to interpret the low-res Pa/Th data with such age uncertainty in this way. I think the authors need to exercise a bit more caution.

**Response 21:**

This interpretation is directly taken from Böhm et al. (2015). We decided to remove the part of the discussion where this interpretation appears (please see response 4 to R1).

L226: Following Landais et al. (2013), we propose that. . .

**Response 22:**

This sentence belongs to the part of text we will remove (please see response 4 to R1).

L242: where does the 70 ppb number come from? Based on the estimated degree of smoothing, what do you think the true magnitude of the DO event was like?

**Response 23:**

All values reported in this manuscript are derived directly from the ice core record and can be interpreted as minimum magnitudes because the firn column acts as a low pass filter. The magnitude of the event in the atmosphere must have been larger than what is recorded in the ice core record. In the frame of the comparison exercise between amplitude and insolation (please see response 1), we plan to deconvolute the 134-ka event. Therefore, we will be able to provide an estimate of the atmospheric magnitude of the event.

L248-249: I agree with this conclusion. Note that during the H1 CH$_4$ event Chinese speleothems get more positive (and more negative for the 134 ka event).

**Response 24:**

L248: The revised text now reads: "Secondly, intra-stadial CH$_4$ variability is also characterized by abrupt CO$_2$ jumps, millennial-scale increase in $\delta^{18}$O$_{atm}$, enrichment in speleothem $\delta^{18}$O(CaCO$_3$) and the absence of concomitant N$_2$O variability (Bauska et al., 2016, 2018; Fischer et al., 2019; Guillevic et al., 2014; Marcott et al., 2014; Rhodes et al., 2015; Schilt et al., 2013, 2010b). The simultaneous occurrence of the CH$_4$ and N$_2$O pulses at $\sim$134 ka BP, the depletion in speleothem $\delta^{18}$O(CaCO$_3$) and the lack of any fluctuation in the $\delta^{18}$O$_{atm}$ record (Landais et al., 2013) contradict these observations (Fig. 3 and 4)".

L255: The magnitude of CH$_4$ rises during DO events is modulated by the NH insolation signal. Based on that, can you place the 134 DO event into context for us? Are we expecting a large or big CH$_4$ DO signal at 134 ka based on insolation?

**Response 25:**

We will investigate this (please see response 1).

L261: strengthening of the ASIAN monsoon system. . .. (probably SH tropics show opposite)

**Response 26:**

L261: The revised sentence now reads: "These data indicate a transient strengthening of the northern hemisphere tropical monsoon systems consistent with a northward shift of the ICTZ."

L285-292: the quenching of the emergent DO event by a MWP pulse does not explain the difference to the Bølling, because the Bølling was coincident with MWP 1A, which seemed not to hurt it!

**Response 27:**

We agree (please see response 1).

L302: As noted above, I think the short duration of the event is fully in line with expectations from DO events of the last glacial cycle, most notably DO2.

**Response 28:**

We agree (please see response 1).

Fig 2 caption: what is the statement about AMOC vigor based on? And why are the transitions not aligned with those in the CH$_4$ data? I think it would make more sense to align the boxes using your data. Also, shouldn't the newly discovered DO event have a red background (interstadial)?

**Response 29**:

We will remove the statement about the vigor of the AMOC. We will also make other modifications to Fig.2 and 3 (please see response 6 to R1).

Fig 4, right panel: why not show the $\delta^{18}O_{atm}$ here also? Would be very helpful. Can you show insolation somewhere, as well as Antarctic temperature? The slowdown in warming at 130.5 ka can be pointed out that way.

**Response 30**:

We will add insolation curves (Berger, 1978) in both panels. We will also add $\delta^{18}O_{atm}$ and Antarctic temperature in the right panel. For Antarctic temperature we will replace the $\delta^{15}N(N_2)$ by the temperature stack of Parrenin et al. (2013).

---

## Author Comment (AC3) · 5 Mar 2021

Dear referees, dear editor

We would like to thank the two anonymous reviewers and the editor for having carefully commented on our manuscript. Please find our response attached as a supplementary file.

Best regards

Loïc Schmidely

Please also note the supplement to this comment:
https://cp.copernicus.org/preprints/cp-2020-131/cp-2020-131-AC3-supplement.pdf